# IDFSR: Personalized Face Super-Resolution with Identity Decoupling and Fitting

## Abstract

In recent years, face super-resolution (FSR) methods have achieved remarkable progress, generally maintaining high image fidelity and identity (ID) consistency under standard settings. However, in extreme degradation scenarios (e.g., scale $> 8\times$), critical attributes and ID information are often severely lost in the input image, making it difficult for conventional models to reconstruct realistic and ID-consistent faces. Existing methods tend to generate hallucinated faces under such conditions, producing restored images lacking authentic ID constraints. To address this challenge, we propose a novel FSR method with Identity Decoupling and Fitting (IDFSR), designed to enhance ID restoration under large scaling factors while mitigating hallucination effects. Our approach involves three key designs: 1) **Masking** the facial region in the low-resolution (LR) image to eliminate unreliable ID cues; 2) **Warping** a reference image to align with the LR input, providing style guidance; 3) Leveraging **ID embeddings** extracted from ground truth (GT) images for fine-grained ID modeling and personalized adaptation. We first pretrain a diffusion-based model to explicitly decouple style and ID by forcing it to reconstruct masked LR face regions using both style and ID embeddings. Subsequently, we freeze most network parameters and perform lightweight finetuning of the ID embedding using a small set of target ID images. This embedding encodes fine-grained facial attributes and precise ID information, significantly improving both ID consistency and perceptual quality. Extensive quantitative evaluations and visual comparisons demonstrate that the proposed IDFSR substantially outperforms existing approaches under extreme degradation, particularly achieving superior performance on ID consistency.

## 1 Introduction

Face super-resolution (FSR) has significant application value in downstream tasks such as identity (ID) recognition and facial analysis (Jiang et al., 2021; Zhang et al., 2020). In scenarios with strict ID requirements, FSR models must not only enhance image quality but also maintain ID consistency. Existing FSR methods are capable of producing high-fidelity, ID-consistent face images under moderate degradation (Chen et al., 2018; Zhou et al., 2022). However, under extreme degradation conditions (e.g., scaling factors exceeding $8\times$), critical ID and attribute information in the input image is often severely lost, turning the reconstruction task into a highly ill-posed problem (Alekseev & Navon, 2001). In the absence of strong prior constraints, the

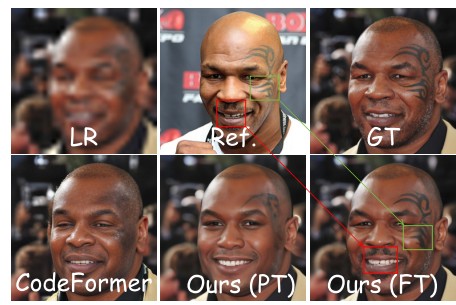

Figure 1: **Visualization of the pretraining (PT) and finetuning (FT) result.** The FT result exhibits significant ID granularity.

solution space becomes underdetermined, with numerous plausible local optima, making the model prone to generating "hallucinated" results that fail to accurately preserve the target ID. For example, as shown in Fig. 1, the conventional method (CodeFormer (Zhou et al., 2022)) may erroneously reconstruct a blurry tattoo region into an unrealistic dark patch.

Reference-based FSR methods (Zhao et al., 2023; Li et al., 2020) provide a promising solution to this problem by leveraging high-quality (HQ) semantic information from reference images to restore missing details in low-resolution (LR) inputs. However, single-reference approaches are highly sensitive to pose, lighting, and expression variations, and typically only perform well when the reference and LR images are closely aligned (Li et al., 2018; Dogan et al., 2019). Although multi-reference methods (Li et al., 2020; Nitzan et al., 2022) attempt to incorporate richer ID features, most existing approaches fail to disentangle ID information between the reference and LR images. **This disentanglement is crucial**: FSR is a pixel-wise reconstruction task, and the LR image itself contains strong structural priors. During training, models tend to over-rely on the residual ID clues in the LR input, thereby neglecting or underutilizing the complementary information provided by the reference images. As a result, the reconstructed faces may resemble the correct ID to some extent, but fail to accurately reproduce finer attributes and details.

To address these challenges, we propose a diffusion-based two-stage framework for FSR via **ID** **D**ecoupling and **F**itting (IDFSR). IDFSR centers around ID-disentangled pretraining and personalized fine-tuning, aiming to achieve high-fidelity reconstruction with consistent ID and attribute information. Specifically, we introduce three key components to obtain an ID-decoupling representation:

- **Corrupted ID Masking**. We roughly detect the facial region in the LR image and apply masking as a strong conditional constraint. This design preserves background consistency and prevents the model from relying on incomplete or erroneous ID cues during training, thereby enhancing its dependency on and utilization of reference information.
- **Style-Conditioned Embedding**. We spatially align the reference face region to the LR face and extract a style embedding via a style encoder. This embedding is integrated into the diffusion model through feature fusion, providing coarse-grained appearance and motion guidance from the reference image.
- **Ground Truth ID Embedding and ID Fitting**. During pretraining, we extract ID embeddings from GT images using a well-trained ID encoder and integrate them into the model via cross-attention. In the fine-tuning, we freeze all model parameters and optimize the ID embeddings using a few samples from the same ID to achieve generalized and precise decoupling representation.

As shown in Fig. 1, the pretraining phase yields reconstructions with basic ID similarity, while ID fitting significantly enhances attribute fidelity and fine-grained details. Extensive experiments across multiple benchmark datasets validate the effectiveness of each proposed component and the framework's ability to suppress hallucination artifacts. Furthermore, comprehensive quantitative comparisons with various SOTA methods show that our approach achieves a significant performance improvement of **20%**, demonstrating the clear superiority of IDFSR in both reconstruction quality and ID consistency.

## 2 RELATED WORKS

**Single-Image Face Super-Resolution.** Different from natural image super-resolution (Zhang et al., 2018b; Dai et al., 2019; Guo et al., 2024; 2025; Dai et al., 2024), face super-resolution (FSR) (Tomar et al., 2023) poses unique challenges due to the need for accurate restoration of fine-grained facial details, ID consistency, and structural priors. Early methods on FSR predominantly relied on direct regression in pixel space; however, empirical studies have shown that such approaches tend to produce overly smoothed results (Huang et al., 2017; Chen et al., 2018). To improve perceptual quality, generative models have been introduced, leveraging prior distributions or explicit dictionaries to enhance the naturalness of reconstructed images (He et al., 2022; Yang et al., 2023). Nonetheless, due to the inherent ambiguity in generative modeling, these methods still face challenges in maintaining pixel-level consistency and ID fidelity. To balance consistency and diversity, recent studies have incorporated conditional generation mechanisms, using facial priors such as landmarks, segmentation maps, and 3D structures as auxiliary constraints to enforce geometric consistency (Yang et al., 2025; Bulat & Tzimiropoulos, 2018). While these approaches have improved geometric reconstruction to some extent, ID preservation remains limited. To address this, ID recognition losses have been employed to enhance ID fidelity (Chen et al., 2020). However, under large degradation factors, ID cues in LR inputs are severely diminished, making models prone to generating "hallucinated" identities.

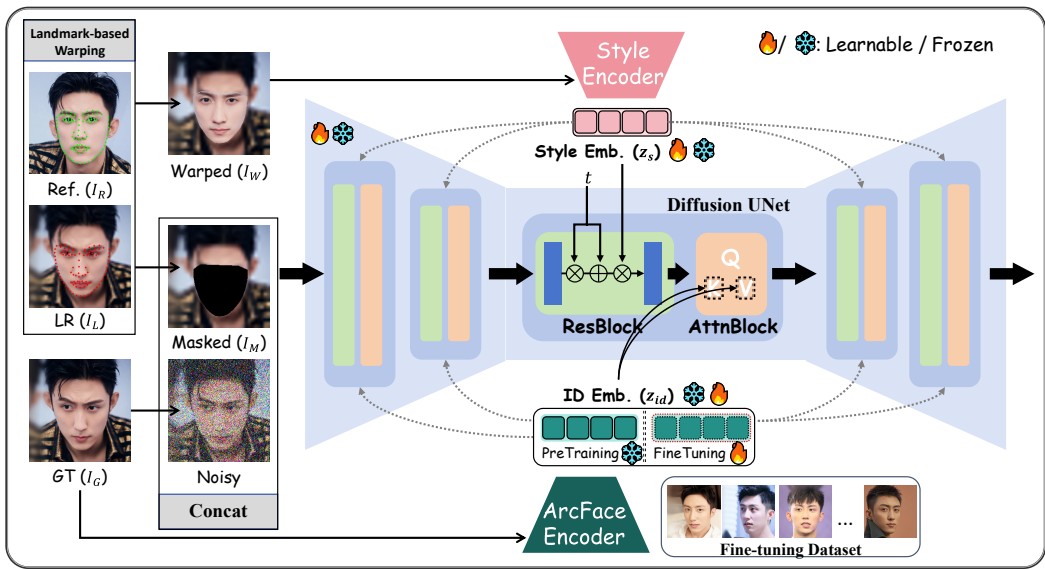

Figure 2: A schematic diagram of single-step diffusion in IDFSR, including input preprocessing and the overall model architecture.

**Reference-Based Face Super-resolution.** To alleviate the severe information loss under high degradation, reference-based FSR methods have been extensively explored. GFRNet (Li et al., 2018) proposed a dual-network architecture—WarpNet aligns the reference image to the LR target via optical flow, while RecNet performs reconstruction. Landmark loss and total variation regularization are adopted to enhance alignment accuracy. However, large pose or expression differences still lead to misalignment issues. GWAINet (Dogan et al., 2019) later introduced ID loss to improve ID consistency but remained heavily dependent on precise alignment. ReFine (Chong et al., 2025) improved alignment using a fine-tuned landmark detector and employed spatial minimality and cycle consistency losses to guide attribute transfer. ASFFNet (Li et al., 2020) selected suitable references via landmark similarity and aligned them in spatial and illumination domains using MLS and AdaIN, followed by multi-stage feature fusion. DMDNet (Zhao et al., 2023) used a dual-memory structure to separately store structural and ID features, enabling adaptive fusion across references. MyStyle (Nitzan et al., 2022) and similar methods construct ID and attribute sets from multiple references. Recently, MGFR (Tao et al., 2024) utilized diffusion models to fuse text, reference images, and ID cues via dual-control adapters and two-stage training for controllable face restoration.

## 3 METHODOLOGY

### 3.1 OVERVIEW

Fig. 2 illustrates the overall architecture of IDFSR. Given a LR image $I_L$, its corresponding GT image $I_G$, and a same-ID reference image $I_R$, IDFSR consists of a parameterized network $\theta$, a style encoder $\mathcal{E}_s$, and a pretrained ArcFace ID encoder $\mathcal{E}_{id}$ (Deng et al., 2019). Prior to training, we perform two preprocessing steps:

- A finetuned face landmark detector is used to mask the facial region of $I_L$, producing the masked image $I_M$.
- A landmark-based warping is applied to map the facial region of $I_R$ onto that of $I_L$, generating the warped image $I_W$.

In the pretraining stage, we adopt Denoising Diffusion Probabilistic Model (DDPM (Ho et al., 2020)) to model the pixel space, using the style embedding $z_s = \mathcal{E}_s(I_W)$ and ID embedding $z_{id} = \mathcal{E}_{id}(I_G)$ as conditions. $I_M$ is concatenated with the noise as input. The simplified loss function at diffusion timestep $t$ is defined as:

$$\mathcal{L}_{\text{sim}} = \mathbb{E}_{x_0, \epsilon | z_s, z_{\text{id}}, I_M} \left\{ \| \epsilon - \epsilon_\theta([x_t, I_M], t, z_s, z_{\text{id}}) \|^2 \right\}. \tag{1}$$

where $[\cdot]$ denotes the concatenation operation, and $x_t$ represents the noisy of the GT image at time step $t$. In the finetuning stage, we freeze the diffusion network $\theta$ and the style encoder $\mathcal{E}_s$, and replace $z_{\text{id}}$ with a trainable embedding vector. This vector is then optimized using a small number of same ID samples, under the same training objective, enabling personalized and accurate ID control. It is worth noting that IDFSR maintains superior performance even when landmark detection is imprecise or reference images are unavailable, effectively compensating for the shortcomings of other methods under such conditions and demonstrating strong robustness.

## 3.2 LANDMARK DETECTION AND WARPING

We use RetinaFace (Deng et al., 2020) to extract facial landmarks from HQ images and finetune the detector to better accommodate LR images. The facial region of the reference image $I_R$ is then aligned to that of the LR image $I_L$ via affine transformation based on the detected landmarks, resulting in a warped image $I_W$. This pseudo-alignment introduces coarse spatial correspondence between the reference and the LR image. We provide the pseudocode of the warping process in the Appendix E.1.

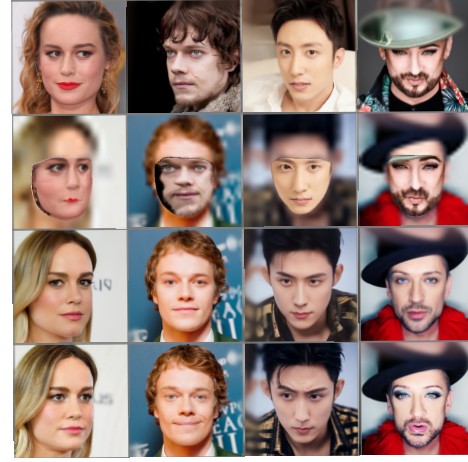

(a) Bad Case     (b) Appropriate Cases

Figure 3: **Warping Analysis**. From top to bottom: reference images, warped images, SR images, and GT images.

However, due to variations in pose or inaccuracies in landmark prediction, such warping is often imperfect and may introduce local distortions or mismatches. Interestingly, we find that this imperfection does not hinder the model's performance—instead, it acts as a form of natural **data augmentation**. The diversity in warped faces encourages the model to avoid over-reliance on precise spatial correspondence and instead to attend to global ID semantics. This drives the diffusion network to rely more heavily on the ID embedding $z_{\text{id}}$ for reconstructing missing facial regions, rather than naively copying textures from the warped reference. As a result, our model becomes inherently robust to variations in reference quality and alignment. As shown in Fig. 3, IDFSR still generating plausible and ID-consistent faces even under severe misalignment.

## 3.3 PRETRAINING AND FINETUNING

Our diffusion backbone follows the U-Net architecture used in Guided Diffusion (Dhariwal & Nichol, 2021), consisting of residual blocks, attention modules, and upsampling/downsampling layers. Our innovative conditional injection method effectively achieves ID decoupling and fitting modeling.

The style encoder adopts the encoder part of the diffusion U-Net. Inspired by DiffAE (Preechakul et al., 2022), we introduce adaptive group normalization (AdaGN (Wu & He, 2018)) to inject style conditions. For a feature map $h \in \mathbb{R}^{c \times h \times w}$, the AdaGN is formulated as:

$$\text{AdaGN}(h, t, z_s) = z_f \cdot (t_s \text{GroupNorm}(h) + t_b), \tag{2}$$

where $z_f \in \mathbb{R}^c$ is obtained via an affine transformation applied to the style embedding $z_s$:

$$z_f = \text{MLP}_{\text{style}}(z_s), \tag{3}$$

and $(t_s, t_b) \in \mathbb{R}^{2 \times c}$ are obtained by applying an MLP to the sinusoidal position embedding $\psi(t)$:

$$(t_s, t_b) = \text{MLP}_{\text{time}}(\psi(t)), \tag{4}$$

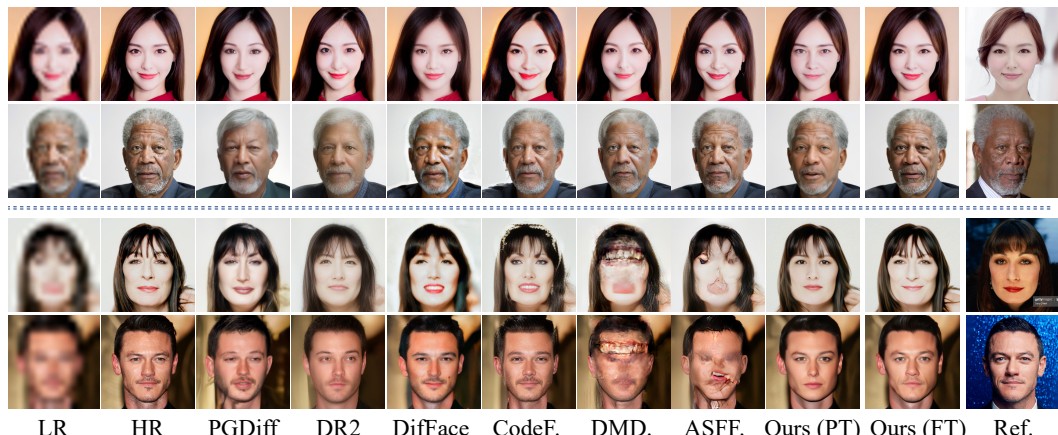

LR HR PGDiff DR2 DifFace CodeF. DMD. ASFF. Ours (PT) Ours (FT) Ref.

Figure 4: Visualization results of different methods under upsampling scales of 8 and 16, separated by double dashed lines. For reference-based methods—DMDNet, ASFFNet, and our IDFSR—a single ID is randomly selected as the reference image. FT and PT refer to finetuning and pretraining methods, respectively. Please zoom in for a better viewing.

In the pretraining, we leverage a pretrained ArcFace encoder (Deng et al., 2019) to extract ID embeddings from the GT images. These ID embeddings are injected into the attention modules via cross-attention mechanisms to provide fine-grained ID priors. The diffusion network and the style encoder are trained from scratch, initialized with DiffAE weights pretrained on the FFHQ dataset (Karras et al., 2019). In the finetuning stage, we freeze both the U-Net and the style encoder, and solely optimize a learnable ID embedding initialized from a reference image. Note that the ArcFace encoder is not required during finetuning. Both stages share the same training hyperparameters: a maximum diffusion step of $T = 1000$, a linear $\beta$ schedule, and DDIM sampling with 20 steps during inference.

## 4 EXPERIMENTS

### 4.1 EXPERIMENTAL SETTING

Our comparison experiments include both general and personalized ones. In the pretraining phase, ID embeddings from reference images are used to achieve general FSR, while in the fine-tuning phase, we focus more on personalized FSR for the same ID. The pretraining strategy is based on the DiffAE model (Preechakul et al., 2022), which was trained for 2 days on four 3090 GPUs, followed by 20 minutes of fine-tuning on a single 3090 GPU with the same ID samples. Detailed implementations are provided in the Appendix E.

**Datasets**. We uniformly sample 10% of the IDs from CelebRef-HQ, with the remaining 90% used for pretraining. Each sampled ID is equally divided into a fine-tuning set and a test set. We filter out certain identities that may contain only a small number of images and additionally select several representative identities to better evaluate ID fitting. Consequently, the training set contains approximately 900 IDs and 9,000 images, while the test set contains 56 IDs and 586 images. Additionally, CASIA-WebFace and the video dataset CelebText are used for generalization evaluation after quality filtering. See the Appendix D for further details.

**Metrics and Baselines.** For evaluation, we adopt metrics, including PSNR, SSIM, LPIPS (Zhang et al., 2018a), FID (Heusel et al., 2017), CLIPIQA (Wang et al., 2023a), MUSIQ (Ke et al., 2021), and ID similarity (IDS) measured by DeepFace (Serengil & Ozpinar, 2020). We compare our method against SOTA reference-free and reference-based approaches, including CodeFormer (Zhou et al., 2022), DR2 (Wang et al., 2023b), DifFace (Yue & Loy, 2024), PGDiff (Yang et al., 2023), DMDNet (Zhao et al., 2023), and ASFFNet (Li et al., 2020).

| Methods | ×8 | | | | | | |
| --- | --- | --- | --- | --- | --- | --- | --- |
| | PSNR↑ | SSIM↑ | LPIPS↓ | FID↓ | CLIPIQA↑ | MUSIQ↑ | IDS↓ |
| CodeFormer | 24.42 | 0.7147 | 0.1489 | 31.88 | 0.6873 | 68.98 | 0.3865 |
| PGDiff | 22.73 | 0.6892 | 0.2223 | 46.05 | 0.5390 | 56.10 | 0.6080 |
| DR2 | 23.83 | 0.7067 | 0.1998 | 44.46 | 0.6307 | 63.35 | 0.5459 |
| DifFace | 24.41 | 0.7255 | 0.1694 | 32.94 | 0.6132 | 60.54 | 0.3815 |
| StableSR | 23.95 | 0.7174 | 0.1523 | 28.34 | 0.6006 | 67.58 | 0.3995 |
| DPI | 24.04 | 0.7564 | 0.1271 | 27.94 | 0.6850 | 70.32 | 0.3342 |
| ASFFNet | 21.82 | 0.6624 | 0.1764 | 26.25 | 0.6158 | 65.81 | 0.3169 |
| DMDNet | 22.36 | 0.6984 | 0.1733 | 29.76 | 0.6570 | 68.28 | 0.4169 |
| Ours (PT) | 24.29 | 0.6996 | 0.1554 | 27.89 | 0.6798 | 69.30 | 0.3173 |
| Ours (FT) | 28.85 | 0.7604 | 0.1031 | 26.69 | 0.7092 | 73.73 | 0.2242 |
| Methods | ×16 | | | | | | |
| | PSNR↑ | SSIM↑ | LPIPS↓ | FID↓ | CLIPIQA↑ | MUSIQ↑ | IDS↓ |
| CodeFormer | 20.72 | 0.5947 | 0.2379 | 43.89 | 0.6834 | 67.85 | 0.6673 |
| PGDiff | 20.47 | 0.6202 | 0.2727 | 55.70 | 0.5008 | 52.19 | 0.7355 |
| DR2 | 22.35 | 0.6356 | 0.2481 | 50.84 | 0.6102 | 62.60 | 0.7082 |
| DifFace | 22.23 | 0.6737 | 0.2195 | 38.65 | 0.5957 | 57.84 | 0.5948 |
| StableSR | 22.65 | 0.6680 | 0.2083 | 37.48 | 0.6818 | 66.82 | 0.6644 |
| DPI | 23.48 | 0.6849 | 0.1997 | 38.35 | 0.6886 | 68.26 | 0.5532 |
| ASFFNet | 18.23 | 0.5732 | 0.2457 | 40.64 | 0.5827 | 63.72 | 0.7955 |
| DMDNet | 17.38 | 0.5590 | 0.2497 | 46.34 | 0.6051 | 64.05 | 0.8291 |
| Ours (PT) | 23.32 | 0.6863 | 0.2062 | 38.58 | 0.6723 | 68.56 | 0.5227 |
| Ours (FT) | 24.09 | 0.7158 | 0.1845 | 35.50 | 0.6992 | 71.83 | 0.3625 |

Table 1: Quantitative comparison on the CelebRef-HQ dataset with upsampling scales of $8\times$ and $16\times$. FT and PT refer to finetuning and pretraining methods, respectively. **Red** and blue indicate the best and the second best.

## 4.2 QUALITATIVE COMPARISONS

As shown in Fig. 13, we present a qualitative comparison of current SOTA FSR methods. Under moderate degradation ($8\times$), non-reference methods are able to achieve high-fidelity image reconstruction, but they still exhibit significant discrepancies in facial consistency and fine-grained detail recovery. Unconditional generative models, such as PGDiff and DR2, tend to generate facial hallucinations, i.e., unrealistic or erroneous facial features. In contrast, conditional generative methods with discriminative constraints, such as DifFace, perform better in pixel-level consistency but still struggle to preserve fine ID features. This limitation primarily arises from the tendency of these methods to learn local average during training, which diminishes their ability to model fine-grained features. Our pretrained model exhibits similar characteristics to some extent, but by incorporating a reference image, it outperforms non-reference methods in alleviating facial hallucinations and improving ID consistency. Meanwhile, SOTA reference-based methods like ASFFNet maintain a better balance between ID consistency and attribute recovery at $8\times$ degradation. However, their performance significantly deteriorates under extreme degradation (such as $16\times$), with their effectiveness heavily reliant on precise matching between the LR and reference images. In contrast, our proposed personalized fine-tuning mechanism significantly enhances the model's ability to preserve ID features and attribute details, demonstrating greater robustness under extreme low-quality conditions. Under the $16\times$ setting, IDFSR achieves notable advantages in both ID consistency and visual fidelity.

## 4.3 QUANTITATIVE COMPARISONS

As shown in Table 1, under moderate degradation, the pretrained model does not achieve optimal performance in pixel-level consistency metrics such as PSNR, SSIM, and LPIPS. However, under severe degradation (e.g., $16\times$), attribute transfer significantly enhances performance, enabling the pretrained model to achieve SOTA results in pixel-level consistency. Further analysis reveals that perceptual quality metrics—including FID, CLIPIQA, and MUSIQ—also demonstrate competitive performance, validating the effectiveness of the pretraining paradigm. As illustrated in Table 1, our personalized fine-tuning strategy achieves remarkable SOTA results across pixel-level consistency,

| Methods | IDV ↑ | ACC$_{Gen}$ ↑ | ACC$_{Emo}$ ↑ | ACC$_{Ra}$ ↑ | Diff$_{Age}$ ↓ |
|---|---|---|---|---|---|
| CodeFormer | 50.3% | 95.2% | 64.3% | 64.8% | $4.6 \pm 4.3$ |
| PGDiff | 32.9% | 88.7% | 51.0% | 68.8% | $5.2 \pm 4.6$ |
| DR2 | 38.9% | 91.9% | 57.6% | 71.5% | $5.2 \pm 5.1$ |
| DiffFace | 73.0% | 95.4% | 67.4% | 77.3% | $4.6 \pm 4.4$ |
| ASFFNet | 21.4% | 72.1% | 39.6% | 45.2% | $7.5 \pm 7.2$ |
| DMDNet | 18.7% | 66.3% | 36.8% | 42.5% | $7.6 \pm 7.3$ |
| **Ours (FT)** | **89.6%** | **98.7%** | **66.2%** | **96.6%** | $\mathbf{3.3 \pm 1.7}$ |

Table 2: The qualitative comparison of face ID verification (IDV) and attribute consistency, including gender accuracy (ACC$_{Gen}$), emotion accuracy (ACC$_{Emo}$), race accuracy (ACC$_{Ra}$), and age difference (Diff$_{Age}$).

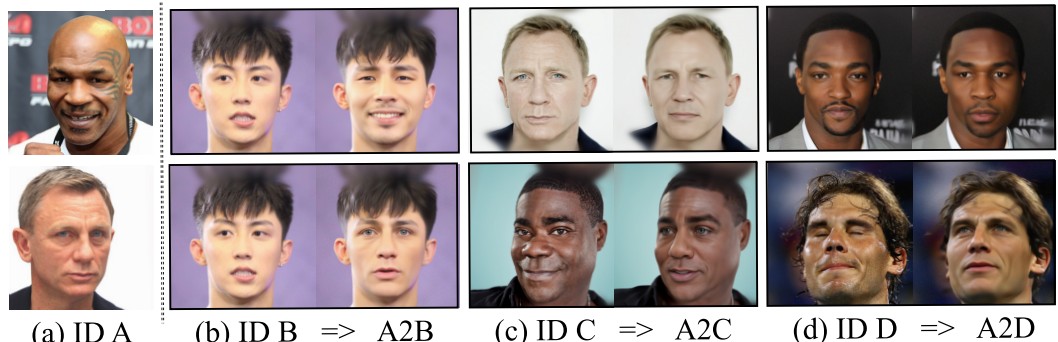

(a) ID A      (b) ID B  =>  A2B      (c) ID C  =>  A2C      (d) ID D  =>  A2D

Figure 5: **Visualization of cross-ID attribute transfer**. We first fit the embedding on ID A, then perform FSR using the LR and style ID from another ID.

perceptual quality, and ID consistency. Specifically, fine-tuning improves pixel-level consistency metrics by approximately 15%, significantly boosts ID consistency, and surpasses other methods while maintaining high visual fidelity.

In fact, there exists a discrepancy in embeddings between the training and testing stages of the pretraining method. Specifically, GT ID embeddings are used during pretraining, whereas reference ID embeddings are employed during testing. This mismatch may lead to model performance being influenced by the fine-grained details of the reference image, potentially resulting in misalignment of attribute features. We will delve deeper into this potential editing effect in the next Section.

### 4.4 FACE VERIFICATION AND ATTRIBUTE ANALYSIS

We further perform extensive evaluations on fine-grained attribute consistency to verify the semantic reconstruction ability of our method. Specifically, we use the DeepFace (Jiang et al., 2021) analysis toolkit to conduct face verification and attribute prediction between the GT and the corresponding SR images. Face verification is based on ID similarity, where a default threshold determines whether the GT and SR images belong to the same person. The attribute prediction includes four dimensions: age, gender, emotion, and race.

We quantitatively compare different methods using recognition accuracy under a $16\times$ scale setting, as shown in Table 2. IDFSR shows a clear advantage in ID verification accuracy. In comparison, many existing methods achieve less than 50% accuracy, indicating significant identity distortion and demonstrating the robustness of our approach under severe degradation. While most methods maintain reasonable gender consistency, emotion and race consistency remain challenging. Our method achieves 96% accuracy in race prediction, which benefits from the use of personalized ID embeddings However, the accuracy for emotion prediction is notably lower. This is mainly because our method does not incorporate strong priors from the LR image space. Instead, it generates faces based on the reference image and customized embeddings, which can affect the expression. We further discuss this issue in the next Section.

# 5 ABLATION STUDY

## 5.1 ID DECOUPLING AND GENERALIZATION VERIFICATION

We design a cross-ID experiment to evaluate the fine-grained modeling capability of the learned ID representation. Specifically, we fit an ID embedding on images of a subject (ID A), and then apply this embedding to restore images of a different subject (ID B). During testing, the embedding of ID A is used in conjunction with LR and reference images from other identities. As shown in Fig. 5, despite the mismatch in ID, the fitted ID A embedding can still effectively modify the target facial attributes, while preserving the background and style. This observation suggests that the ID embedding is decoupled from the LR input and style encoding, indicating that it is learned independently of identity-aligned pixel supervision or image warping processes. Moreover, the success of the cross-ID experiment further demonstrates the strong generalization capability of the fitted ID representation.

## 5.2 LATENT EDITING EFFECTS OF STYLE EMBEDDING

The style embedding $z_s$ serves as the only explicitly designed pathway in our framework for capturing spatial information of the face, although the warping operation is not always perfectly aligned. In Methodology, we have discussed the impact of warping alignment; here, we further explore the potential editing effect of style embedding in the context of FSR.

As shown in Fig. 6, we select three images of the same ID with variations in lighting, makeup, and facial expressions. Two of these, which exhibit large differences in facial motion, are used as reference images under $16\times$ downsampling. It can be clearly observed that the SR outputs exhibit a noticeable resemblance in facial expression to the reference images. Despite certain misalignments at the pixel level, our method achieves high ID consistency and visual fidelity. In contrast, most existing approaches prioritize pixel-level alignment, which restricts the flexibility of ID generation. Notably, aside from variations in facial expressions, the differences in background and fine details are minimal. This further validates the effectiveness of our key design in decoupling ID features.

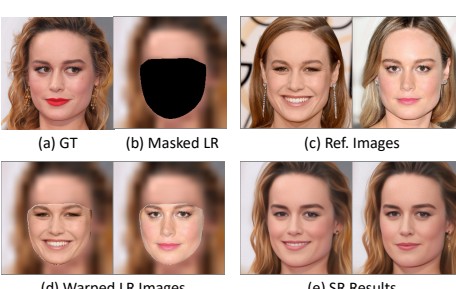

(a) GT  (b) Masked LR  (c) Ref. Images

(d) Warped LR Images  (e) SR Results

Figure 6: **Latent Editing Effects of Style Embedding.** Using different styles of reference images usually only changes the action features, while the attributes and background features have a high degree of consistency.

## 5.3 THE IMPACT OF REFERENCE IMAGE QUANTITY

To investigate the impact of the number of reference images on fine-tuning performance, we select samples from the CelebHQ-Ref test set, where each ID contains more than 10 images. Specifically, we gradually increase the number of reference images used for fine-tuning, ranging from 2 to 10, and evaluate the performance using three normalized metrics: pixel-level, ID and attribute consistency. As shown in Fig. 7, we report the average values and confidence intervals of these metrics across 10 identities.

The overall trend shows that increasing the number of reference images helps improve the model's consistency across all evaluation metrics. In the early stages, introducing a small number of images brings limited improvement, which may be due to the pretrained model already possessing a certain level of single-image reference capability. The additional images provide limited information in terms of attribute diversity or fine-detail supplementation, especially when there are variations in image quality or pose. Notably, when the number of reference images reaches around five, the performance and stability tend to reach an optimal balance.

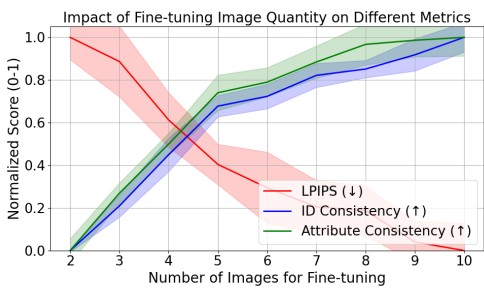 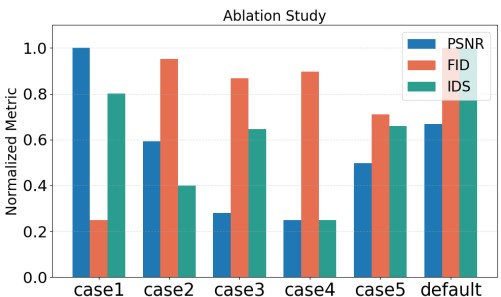

Figure 7: **Effect of Reference Image Quantity on Finetuning Performance.** Results show that performance generally improves with more references, with diminishing returns beyond five images, where performance and stability reach a balance.

Figure 8: **Ablation Study on Model Components.** Since the metrics have different scales, we normalize them to better illustrate the impact of each component on various performance measures.

### 5.4 COMPONENT ANALYSIS

Starting from an unconditional diffusion model, we progressively introduce key components to systematically analyze the contribution of each element to the final performance, as illustrated in Fig. 8. The most basic yet effective diffusion-based super-resolution strategy concatenates the low-resolution (LR) image with the noise input at each timestep (case1), as exemplified by SR3 (Saharia et al., 2022) and SRDiff (Li et al., 2022). This strategy imposes strong constraints that improve generation consistency, but suffers from reduced training efficiency and reconstruction fidelity under severe degradation.

In case2, the LR image is masked before concatenation, effectively transforming the task into an inpainting problem. This introduces greater generative uncertainty. Nevertheless, such strong prior constraints remain crucial for low-level vision tasks. Compared to methods relying on feature-based conditioning or attention mechanisms—such as DiffAE and Stable Diffusion (Rombach et al., 2022) (case3)—this approach offers superior pixel-level consistency but performs worse in terms of perceptual quality. As shown in Fig. 8, case1 achieves the best consistency scores but significantly degrades perceptual quality, while case2 exhibits the opposite trend. Case3 yields moderate performance across all metrics, indicating that it tends to produce structurally plausible but not precisely aligned results.

When the ID embedding is removed under the default setting (case4), we observe that the model tends to generate ID-inconsistent faces due to misalignments introduced by the warping operation, resulting in decreased ID consistency. Removing the style embedding (case5) leads to reconstructions that often lack realistic ID traits, suggesting that ID and style embeddings play complementary roles in preserving ID fidelity. The default configuration effectively integrates the advantages of each component: the masking operation provides a strong background constraint prior, while the ID and style embeddings enable disentangled modeling and accurate fitting of ID and facial attributes. Consequently, the model achieves strong performance in both ID consistency and attribute preservation.

## 6 CONCLUSION

In this paper, we propose IDFSR, a two-stage diffusion-based face super-resolution framework. During the pretraining stage, we achieve facial disentanglement through three key designs, while the finetuning stage enables the fitting of personalized attributes. IDFSR is both generalizable and customizable, demonstrating impressive identity and attribute consistency in both quantitative and qualitative evaluations. Furthermore, extensive ablation studies validate the disentanglement and editability capabilities of our method.

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

## A    DISCUSSION AND LIMITATIONS

**Discussion**: Reference-based super-resolution (Ref-SR) methods often face practical limitations due to the availability and quality of the reference images. As a result, such methods are typically more suitable for specific scenarios, such as when a user's photo album contains a large number of high-quality images of the same identity (ID), when high-resolution (HR) image resources are accessible in enterprise databases, or in video sequences where inter-frame quality discrepancies exist. Our method is designed with a high degree of flexibility, enabling it to remain robust across many such specialized scenarios. In the Appendix, we further discuss the limitations of our approach and explore its extension to reference-free settings, real-world generalization, and video-based applications, highlighting the superiority of our method over existing alternatives.

**Limitations**: The proposed IDFSR pre-trained model can be regarded as a general-purpose single-reference image super-resolution method. However, as evidenced by the visual results, the model tends to learn locally averaged features due to generalization challenges during training, resulting in blurry details and overly smooth attribute representation. To address this issue, we introduce a customized fine-tuning strategy, which significantly enhances ID consistency and the fidelity of attribute details, while effectively mitigating detail loss. It is important to note, however, that this approach relies on images of the same ID and requires a small number of samples as well as a certain amount of fine-tuning time. Moreover, our method primarily targets the problem of non-deterministic reconstruction under severe degradation, as well as model hallucination. In contrast, for cases with mild degradation, existing methods already perform well, and thus our framework is not specifically tailored for such general scenarios. For example, although our proposed mask-decoupling strategy may impact pixel-level consistency, we prioritize perceptual quality, ID consistency, and semantic similarity at the feature level.

Due to the limitations of the masking mechanism, the method lacks strong pixel-level facial priors. Although ID embeddings provide fine-grained attribute information, we observe that for certain non-generic facial attributes—such as gaze direction, accessories, and makeup—the model may fail to faithfully preserve the details present in the low-resolution input. As illustrated in Fig. 6 of the main text, even when the ID and background are consistent, the gaze direction can still exhibit noticeable deviation. Finally, **it is worth discussing whether the proposed method remains effective in scenarios where no usable reference images are available**. We provide a detailed discussion of this issue in the following Section A.C.

## B    BACKGROUND

Diffusion Probabilistic Models (or simply diffusion models) have emerged as a powerful class of generative models (Ho et al., 2020; Rombach et al., 2022), demonstrating impressive performance in tasks such as image synthesis, restoration, and super-resolution. These models are based on a parameterized Markovian process that gradually adds noise to data through a forward process and learns to reverse this process to recover high-fidelity samples.

**Forward Diffusion Process.**    Given a clean image $\mathbf{x}_0$, the forward process perturbs it over $T$ time steps by adding Gaussian noise at each step. The process is defined as:

$$q(\mathbf{x}_t|\mathbf{x}_{t-1}) = \mathcal{N}(\mathbf{x}_t; \sqrt{1-\beta_t}\mathbf{x}_{t-1}, \beta_t\mathbf{I}), \tag{5}$$

where $\beta_t \in (0,1)$ is a variance schedule, often chosen to increase linearly or with a cosine profile. The marginal distribution at arbitrary timestep $t$ can be derived in closed form as:

$$q(\mathbf{x}_t|\mathbf{x}_0) = \mathcal{N}\left(\mathbf{x}_t; \sqrt{\bar{\alpha}_t}\mathbf{x}_0, (1-\bar{\alpha}_t)\mathbf{I}\right),$$
$$\text{where} \quad \bar{\alpha}_t = \prod_{s=1}^{t}(1-\beta_s). \tag{6}$$

This formulation enables efficient sampling of noised data from any intermediate timestep without explicitly simulating all previous steps.

(a) GT      (b) Maked LR    (c) SR w/o Ref.

Figure 9: Our visualization results for FSR without reference images demonstrate the robustness of our method under this setting. In this case, our approach can be viewed as an ID-aware inpainting method, capable of generating plausible results even in the absence of reference guidance. Although the ID consistency is naturally inferior compared to reference-based settings, and the fine-tuning stage cannot fully exploit its ID adaptation capabilities, this experiment highlights the superiority of our method in terms of flexibility and generalization.

**Reverse Denoising Process.** The generative model aims to learn the reverse process $p_\theta(\mathbf{x}_{t-1}|\mathbf{x}_t)$ that gradually denoises $\mathbf{x}_T \sim \mathcal{N}(\mathbf{0}, \mathbf{I})$ to recover $\mathbf{x}_0$. It is modeled as a Gaussian with learnable mean and (often fixed) variance:

$$p_\theta(\mathbf{x}_{t-1}|\mathbf{x}_t) = \mathcal{N}(\mathbf{x}_{t-1}; \boldsymbol{\mu}_\theta(\mathbf{x}_t, t), \boldsymbol{\Sigma}_\theta(\mathbf{x}_t, t)). \tag{7}$$

In the Denoising Diffusion Probabilistic Model (DDPM (Ho et al., 2020)), $\boldsymbol{\mu}_\theta$ is derived from a noise prediction network $\boldsymbol{\epsilon}_\theta$, leading to:

$$\boldsymbol{\mu}_\theta(\mathbf{x}_t, t) = \frac{1}{\sqrt{\alpha_t}} \left( \mathbf{x}_t - \frac{\beta_t}{\sqrt{1 - \bar{\alpha}_t}} \boldsymbol{\epsilon}_\theta(\mathbf{x}_t, t) \right). \tag{8}$$

The model is trained by minimizing the variational bound, which can be simplified into a reweighted noise prediction objective:

$$\mathcal{L}_{\text{simple}} = \mathbb{E}_{t, \mathbf{x}_0, \boldsymbol{\epsilon}} \left[ \left\| \boldsymbol{\epsilon} - \boldsymbol{\epsilon}_\theta \left( \sqrt{\bar{\alpha}_t} \mathbf{x}_0 + \sqrt{1 - \bar{\alpha}_t} \boldsymbol{\epsilon}, t \right) \right\|^2 \right]. \tag{9}$$

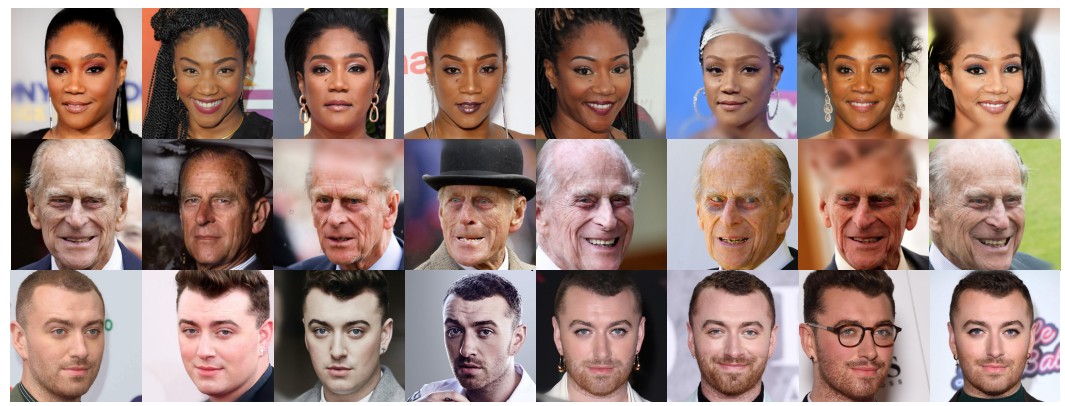

Figure 10: **Visualization of CelebRef-HQ dataset samples.**

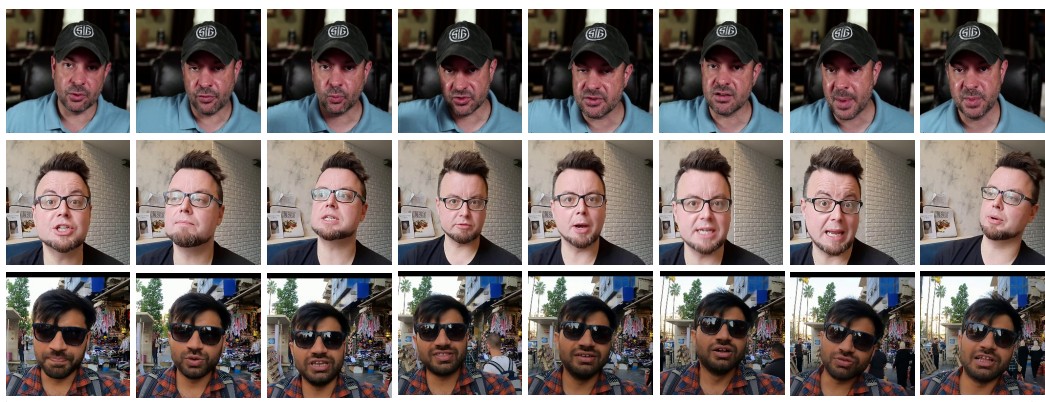

Figure 11: **Visualization of CelebV-Text dataset samples.**

## C EXTENDING IDFSR TO REFERENCE-FREE FSR

Most reference-based SR methods heavily rely on the availability and quality of reference images—for example, they typically require precise alignment through warping or depend on multiple high-quality reference images; otherwise, their performance degrades significantly or they may even fail to work in the absence of reference images. In this Section, we explore how our method can be adapted to the single-image SR scenario without any external references.

We treat the low-resolution (LR) image itself as the reference. Specifically, we use the default pretrained model as the backbone network, extracting style and ID embeddings from the LR image as conditioning inputs, while feeding the masked LR image directly as input without modification. As shown in Fig. 9, even in the absence of any reference image and under severe degradation, IDFSR is still able to reconstruct high-quality facial images effectively. However, due to the lack of strong style and ID priors, the ID consistency cannot be fully preserved in the reference-free setting. As shown in Table 3 (w/o Ref.), quantitative comparisons demonstrate that IDFSR achieves competitive overall performance without using any reference images.

We attribute this to two key factors. First, the backbone model benefits from pretraining with misaligned warping-based data augmentation, enabling the style encoder to tolerate LR inputs without significant performance degradation. Second, the use of LR images as inputs to the ID encoder does not substantially affect performance due to the robustness and reliability of the ID embedding space. In other words, ID embeddings derived from LR images are generally contained within the true ID embedding space, preserving crucial ID information without compromising the reconstruction capability of the overall framework.

| Methods | ×4 | | | | | | |
|---------|------|------|--------|-------|----------|--------|------|
|  | PSNR↑ | SSIM↑ | LPIPS↓ | FID↓ | CLIPIQA↑ | MUSIQ↑ | IDS↓ |
| CodeFormer | 27.74 | 0.8182 | 0.0838 | 24.90 | 0.6893 | 68.99 | 0.2780 |
| PGDiff | 26.33 | 0.7987 | 0.1643 | 33.72 | 0.6694 | 65.72 | 0.3050 |
| DR2 | 26.43 | 0.7932 | 0.1769 | 34.34 | 0.6604 | 65.56 | 0.3338 |
| DifFace | 27.21 | 0.8022 | 0.1172 | 27.67 | 0.6885 | 69.14 | 0.2811 |
| ASFFNet | 26.86 | 0.7941 | 0.1224 | 29.87 | 0.6828 | 67.74 | 0.2342 |
| DMDNet | 26.46 | 0.7943 | 0.1313 | 30.87 | 0.6775 | 67.49 | 0.2420 |
| Ours (PT) | 25.96 | 0.7789 | 0.1576 | 29.83 | 0.6665 | 65.83 | 0.2659 |
| Ours (FT) | 27.66 | 0.8176 | 0.0923 | 25.87 | 0.6937 | 68.87 | 0.1983 |

| Methods | ×32 | | | | | | |
|---------|------|------|--------|-------|----------|--------|------|
|  | PSNR↑ | SSIM↑ | LPIPS↓ | FID↓ | CLIPIQA↑ | MUSIQ↑ | IDS↓ |
| CodeFormer | 18.13 | 0.5308 | 0.4562 | 119.86 | 0.6470 | 53.95 | 0.8669 |
| PGDiff | 18.26 | 0.5431 | 0.4095 | 89.76 | 0.5723 | 53.45 | 0.8549 |
| DR2 | 17.64 | 0.5234 | 0.4879 | 98.94 | 0.5320 | 51.44 | 0.8898 |
| DifFace | 18.86 | 0.5728 | 0.3352 | 73.18 | 0.5604 | 55.15 | 0.8210 |
| ASFFNet | – | – | – | – | – | – | – |
| DMDNet | – | – | – | – | – | – | – |
| Ours (w/o Ref.) | 18.36 | 0.5679 | 0.3654 | 73.41 | 0.5654 | 55.77 | 0.6642 |
| Ours (PT) | 19.95 | 0.6593 | 0.2984 | 63.09 | 0.6327 | 59.64 | 0.5133 |
| Ours (FT) | 20.76 | 0.6675 | 0.2786 | 57.57 | 0.6761 | 62.36 | 0.4773 |

Table 3: Quantitative comparison on the CelebRef-HQ dataset with upsampling scales of 4× and 32×. FT, PT and w/o Ref. refer to finetuning, pretraining and without reference methods, respectively. Red and blue indicate the best and the second best.

Listing 1: TPS warping using Piecewise Affine Transform

```python
def tps_warp_image(src_img, src_points, dst_points, dst_shape):
    from skimage.transform import PiecewiseAffineTransform, warp

    """
    Perform image warping using Piecewise Affine Transform.
    Maps src_points -> dst_points.

    Args:
        src_img     : Source image (HWC).
        src_points  : Source control points, shape (N, 2).
        dst_points  : Destination control points, shape (N, 2).
        dst_shape   : Target image shape (height, width).

    Returns:
        Warped image with shape = dst_shape.
    """

    # Step 1: Estimate piecewise affine transform
    tform = PiecewiseAffineTransform()
    tform.estimate(dst_points, src_points)  # Note: warp uses inverse
        mapping

    # Step 2: Apply warping using the estimated transform
    warped = warp(src_img, tform, output_shape=dst_shape, mode='edge')

    # Step 3: Convert to 0-255 uint8 image
    warped = (warped * 255).astype(np.uint8)

    return warped
```

# D  DATASET INTRODUCTION

**CelebVRef-HQ dataset** (Zhao et al., 2023): comprises 1,005 celebrity identities with a total of 10,555 images, averaging between 3 to 21 images per ID. As illustrated in Fig. 2, images of the

Table 4: Network architecture of our IDFSR based on the DiffAE (Preechakul et al., 2022) and improved DPM architecture (Dhariwal & Nichol, 2021)

| Parameter | IDFSR Architecture |
|---|---|
| Batch size | 24 |
| Base channels | 128 |
| Channel multipliers | [1,1,2,2,4,4] |
| Attention resolution | [16] |
| Style Encoder base ch | 128 |
| Style Enc. attn. resolution | [16] |
| Style Encoder ch. mult. | [1,1,2,2,4,4,4] |
| ID Encoder | Pretrained ArcFace |
| $z_s$ size | 512 |
| $z_{id}$ size | 512 |
| $\beta$ scheduler | Linear |
| Learning rate | $1 \times 10^{-4}$ |
| Optimizer | Adam (no weight decay) |
| Training T | 1000 |
| Diffusion loss | MSE with noise prediction $\epsilon$ |

| | CASIA-WebFace Dataset | | | | CelebV-Text Dataset | | | | | | |
|---|---|---|---|---|---|---|---|---|---|---|---|
| Methods | FID↓ | CLIPIQA↑ | MUSIQ↑ | IDS↓ | PSNR↑ | SSIM↑ | LPIPS↓ | FID↓ | CLIPIQA↑ | MUSIQ↑ | IDS↓ |
| CodeFormer | 51.26 | 0.5793 | 58.27 | 0.7876 | 21.69 | 0.6437 | 0.2085 | 39.93 | 0.6844 | 68.95 | 0.6386 |
| PGDiff | 45.32 | 0.6063 | 65.72 | 0.6900 | 20.97 | 0.6332 | 0.2320 | 47.89 | 0.5480 | 56.30 | 0.7121 |
| DR2 | 42.33 | 0.6230 | 65.32 | 0.6467 | 22.46 | 0.6748 | 0.2119 | 41.22 | 0.6570 | 65.32 | 0.6996 |
| DifFace | 58.39 | 0.5764 | 59.84 | 0.7960 | 22.61 | 0.7018 | 0.1897 | 37.32 | 0.6449 | 64.30 | 0.5982 |
| ASFFNet | – | – | – | – | 18.70 | 0.6181 | 0.2984 | 53.62 | 0.6592 | 64.76 | 0.7765 |
| DMDNet | – | – | – | – | 19.18 | 0.6242 | 0.2765 | 52.61 | 0.6648 | 66.96 | 0.7973 |
| Ours (FT) | 39.55 | 0.6469 | 67.03 | 0.5090 | 26.52 | 0.7774 | 0.1513 | 28.54 | 0.7042 | 71.07 | 0.3227 |

Table 5: Quantitative comparison on the CASIA-WebFace dataset and CelebV-Text dataset with upsampling scales 16×. **Red** and blue indicates the best and the second best.

same ID exhibit significant variations in makeup, clothing, pose, expression, and lighting, which highlights the importance of the proposed method's ability to decouple these entangled factors.

**CASIA WebFace dataset** (Yi et al., 2014): contains 10,575 identities and a total of 494,414 face-cropped images. Since the images are collected from the web, they vary significantly in quality, and the number of reference images per ID is highly imbalanced. To ensure data quality, we employ the quality assessment network proposed by Su et al. (Su et al., 2020) to filter out low-quality images and further remove identities with fewer than two remaining images. The final curated version of this dataset contains approximately 1,035 high-quality identities with a total of 14,908 images. Similar to CelebVRef-HQ, this dataset exhibits variations in pose, makeup, and other facial attributes.

**CelebV-Text dataset** (Yu et al., 2023): is introduced by Yu et al. as the first large-scale face video-text paired dataset designed to support tasks such as natural language-driven facial video generation and retrieval. It contains over 70,000 high-quality facial video clips with a total duration of approximately 279 hours, covering a diverse range of identities, expressions, actions, lighting conditions, and camera angles. This dataset serves as a strong benchmark for research in semantics-guided facial video synthesis and video-text alignment. Due to its large scale, we select only the first five clips from the 69 officially released slices. We then perform quality filtering to obtain the top 1,000 high-quality ID tracks within these slices. As the video duration varies across identities, we extract all frames and uniformly sample 10 images per ID by taking one every 20 frames. As shown in Fig. 3, sample frames (unaligned) illustrate that while the ID and appearance remain largely consistent across frames, the actions vary due to the temporal continuity of video data.

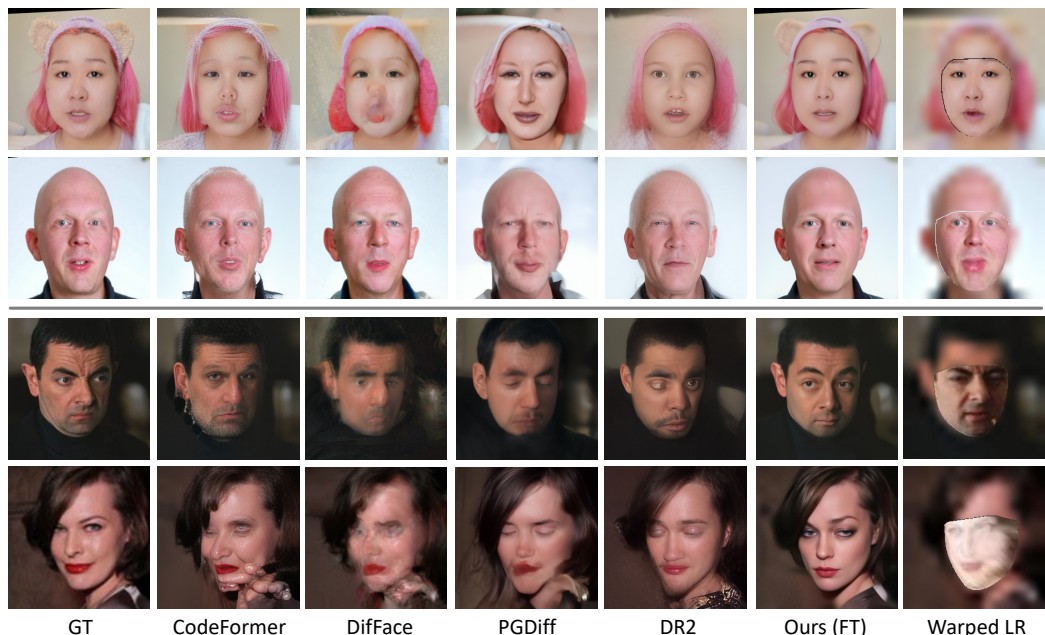

GT      CodeFormer      DifFace      PGDiff      DR2      Ours (FT)      Warped LR

Figure 12: Qualitative visual analysis on the CelebV-Text and CASIA-WebFace datasets. The upper part presents results on CelebV-Text, while the lower part shows results on CASIA-WebFace. Reference-based methods such as DMDNet and ASSFNet tend to fail in most real-world scenarios.

# E MORE IMPLEMENTATION DETAILS

## E.1 WAPRING PROCESS

As shown in the pseudocode of List 1, the function implements spatial image deformation based on Piecewise Affine Transform. Specifically, it estimates a local affine transformation model using control points from the source image (src_points) and their corresponding points in the target image (dst_points). The transformation is performed via inverse mapping, which ensures correct sampling by mapping target coordinates back to source coordinates. Subsequently, the estimated transformation is applied to warp the source image, producing a deformed image that conforms to the target shape dimensions. The final output is normalized.

There exist higher-precision warping methods, such as Thin Plate Spline (TPS) transformation; however, we found that TPS is computationally expensive, especially considering that the warping operation participates in the training process. Moreover, we verified that the correctness of the warping has minimal impact on overall performance.

## E.2 MODEL CONFIGURATION

Our main framework adopts the architecture and parameters of DiffAE, as shown in Table 4. We innovatively introduce three key design modifications, resulting in slight adjustments to the network structure. First, we change the original 3-channel input (single image) to 6 channels by concatenating noise and masked LR inputs. Second, all self-attention mechanisms are replaced with cross-attention mechanisms to incorporate ID embeddings. Other parameters and structures remain unchanged; therefore, we initialize the U-Net and style encoder with the pretrained weights of DiffAE before training. We train the model in parallel using four NVIDIA 3090 GPUs, with a batch size of 24 per GPU. In fact, except for a negligible increase in the number of convolutional filters in the input layer, our method does not introduce any additional parameters. This results in a time complexity nearly identical to current diffusion (DDPM)-based SR methods (Yang et al., 2023; Yue & Loy, 2024; Wang et al., 2023b).

Table 6: **Comparison of Complexity Analysis**: All experiments ard conducted on a single NVIDIA RTX 3090 GPU.

| Method | PGDiff | DR2 | DifFace | StableSR | DPI | IDFSR |
|---|---|---|---|---|---|---|
| Params (M) | 159.7 | 179.31 | 175.42 | 918.93 | 145.53 | 160.7 |
| FLOPs (G) | 185.95 | 918.85 | 272.67 | 2000+ | 136.57 | 168.68 |
| NFEs | 100 | 100 | 100 | 50 | 20 | 20 |
| Single-step Time (ms) | 71 | 89 | 70 | 362 | 50 | 54 |
| Total Inference Time (s) | 7 | 9 | 7 | 18 | 1 | 1 |

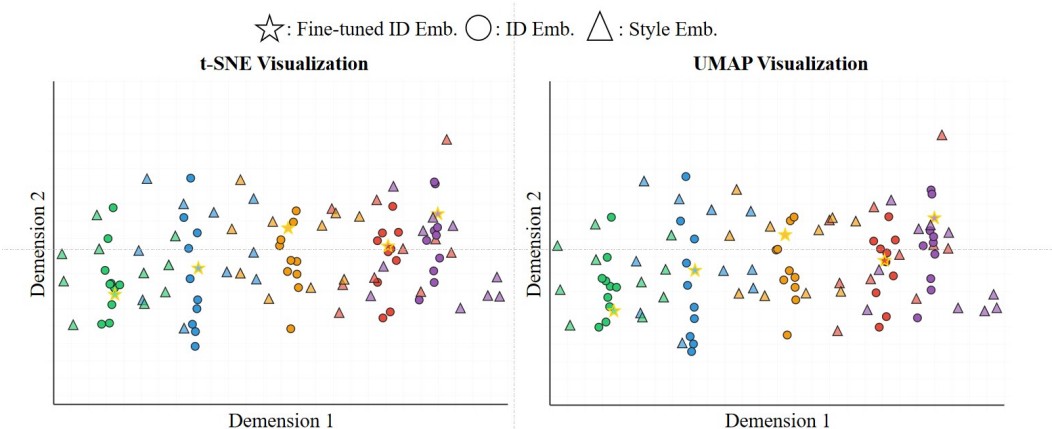

Figure 13: t-SNE and UMAP visualization of embeddings for five identities, with ten images per identity. Each color corresponds to a different identity.

### E.3 COMPLEXITY ANALYSIS

IDFSR, while maintaining a moderate number of parameters and FLOPs, requires only 20 sampling steps to complete inference, with a total runtime of just 1s. In comparison, PGDiff, DR2, and Dif-Face require 100 steps (taking 7–9s), and StableSR takes 18s, demonstrating a significant efficiency improvement. Compared with DPI, IDFSR slightly increases the per-step time (54ms vs. 50ms) but, by leveraging an additional style encoder and a 512-dimensional ID embedding, it efficiently preserves identity and style features, achieving high-quality super-resolution reconstruction, whereas other methods incur higher computational costs.

### E.4 LANDMARK DETECTOR

To detect coarse landmarks in LR images, we follow the ReFine approach and fine-tune RetinaFace. Specifically, for the CelebRef-HQ dataset, we generate LR–GT landmark pairs by applying random scaling to HR images within the training set. For real-world datasets, LR inputs are synthesized using the degradation model from Real-ESRGAN. All training settings follow those of the original RetinaFace. We fine-tune the model starting from the official pretrained weights, using LR images as inputs. Training is performed for 20,000 steps on a single RTX 3090 GPU with a batch size of 4.

### E.5 THEORETICAL ANALYSIS OF DISENTANGLEMENT

Our Fig. 5 and 6 provide initial empirical evidence supporting the independent roles and disentanglement of the style and identity codes. To further validate this phenomenon from a more theoretical perspective, we first employ Mutual Information Neural Estimation (MINE) to quantify the mutual information between the identity representation $z_{id}$ and the style representation $z_s$. Specifically, using ten images from the same identity, we obtain a mutual information of MI = 0.186 nats between the style and identity embeddings. The mutual information within identity embeddings and within

style embeddings is MI = 2.15 and MI = 0.95 nats, respectively, indicating that the two representations are statistically close to independent.

Next, we extract $z_{id}$ and $z_s$ from 1,000 test samples and compute their canonical correlation coefficients (CCA). The average canonical correlation is below 0.17, further confirming the weak dependence between identity and style from a linear correlation perspective.

Finally, we select five identities and randomly sample ten images for each identity to perform a systematic visualization of the embedding space. As shown in the Fig. **??**, both t-SNE and UMAP projections reveal strong intra-class compactness and inter-class separability for the identity embeddings. Different identities form clear and distinguishable clusters in the low-dimensional space, demonstrating the discriminative power of $z_{id}$. Meanwhile, style embeddings from the same identity are dispersed around the corresponding identity cluster, indicating successful disentanglement between identity and style. Moreover, the fine-tuned identity embeddings remain stably concentrated within their respective clusters, suggesting that the fine-tuning process enhances identity consistency without disrupting the global cluster structure.

# F  MORE EXPERIMENTS

## F.1  MILD AND SEVERE SCENARIOS FSR

We quantitatively compare the performance of our method with several existing approaches on the CelebRef-HQ dataset under 4× and 32× scaling scenarios, as shown in Table 3. While our method may not achieve the best performance under mild degradation due to certain limitations, it consistently achieves state-of-the-art (SOTA) performance under severe degradation scenarios, both in pretraining and finetuning settings.

## F.2  VIDEO SCENE FSR

We conduct 16× SR experiments on the CelebV-Text dataset. A pre-trained model is fine-tuned on each video ID. During testing, reference images are randomly selected with a minimum interval of 20 frames. Several interesting observations were made. As shown in the upper part of Fig. 12, due to the high similarity in background across video frames and the variations mainly occurring in the subject's motion, our warping module achieves highly accurate alignment, resulting in superior visual quality and remarkable temporal consistency. Moreover, the quantitative results in Table 5 demonstrate the robustness and effectiveness of our method in video-based scenarios, significantly outperforming other approaches across all evaluation metrics.

## F.3  REAL-WORLD FSR

The CASIA-WebFace dataset is collected from the internet. Although we applied a degree of quality filtering, some real-world noise remains in the images. To simulate realistic scenarios, we further downsampled the images to a resolution of 16×16. During inference, reference images are selected from the same ID. As illustrated in the lower part of Fig. 12, real-world noise is evident in some ground-truth (GT) images, and the warping is less accurate compared to the video-based setting. Nevertheless, our method, with its high flexibility, can effectively capture ID-specific features while leveraging stylistic information from reference images. The results show that our approach achieves superior ID consistency and higher fidelity compared to other methods under real-world conditions. Furthermore, the no-reference metrics in Table 5 also validate the advantage of our method.

## F.4  PROGRESSIVELY DEGRADED THE REFERENCE IMAGE

Following the real-world degradation process defined in Real-ESRGAN Wang et al. (2021), we investigate the robustness of IDFSR under progressively deteriorated reference images. Leveraging the controllable progression of this degradation model, we apply increasingly severe degradations to the reference image until its identity cues vanish completely, corresponding to an extreme "reference-free" scenario.

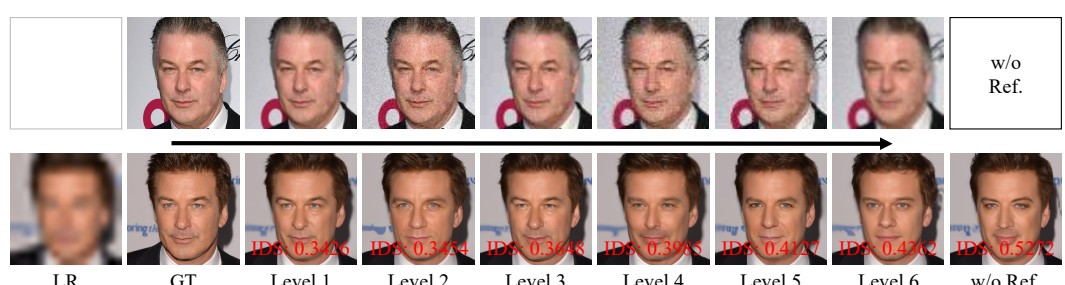

LR    GT    Level 1    Level 2    Level 3    Level 4    Level 5    Level 6    w/o Ref.

Figure 14: **Robustness analysis under progressively degraded reference images.**

$$I_L = \left\{ \left[ (I_H * g_{s,\sigma}) \downarrow_r + \eta_\delta \right]_{\text{JPEG}(q)} \right\} \uparrow_r, \tag{10}$$

where $g_{s,\sigma}$ denotes a Gaussian smoothing kernel with spatial size $s$ and standard deviation $\sigma$, $\eta_\delta$ represents additive Gaussian noise with variance $\delta$, $\text{JPEG}(q)$ applies lossy JPEG compression with quality factor $q$, and $r$ specifies the scale factor used in the downsampling and subsequent upsampling operations.

As shown in Fig. 14, our observations show that under mild degradation, the model performance remains largely unaffected, and the consistency between the generated results and the reference image is well preserved. As the degradation becomes more severe and the reference image loses meaningful identity information, the model gradually degenerates into an identity-aware super-resolution method that relies solely on identity priors. In this extreme case, the model still maintains certain fine-grained identity-related attributes—such as eye color or nose shape inferred from the learned identity embedding—whereas the overall identity consistency inevitably degrades.

