# OpenReview forum: "IDFSR: Personalized Face Super-Resolution with Identity Decoupling and Fitting"
_ICLR.cc/2026/Conference — ICLR 2026 Conference Withdrawn Submission_

### Official Review · Reviewer_1Cgt · 2025-10-27

**Soundness:** 2
**Presentation:** 2
**Contribution:** 2
**Rating:** 2
**Confidence:** 5

**Summary:**

The paper proposes IDFSR, a diffusion-based two-stage framework that aims to improve identity consistency under extreme degradation scenarios. In the first stage, the authors pretrain a diffusion model that reconstructs masked low-resolution faces conditioned on a warped reference image and ground-truth ID embedding. In the second stage, the model performs lightweight fine-tuning to personalize ID embeddings using a few samples from the same identity. The approach claims to decouple style and ID representations, improving robustness and reducing hallucinations in high-scale super-resolution tasks. Extensive experiments on several datasets show improved ID consistency and visual quality over prior works.

**Strengths:**

- The paper identifies a realistic problem of maintaining ID consistency under extreme degradation, and provides a well-structured solution using diffusion-based modeling and personalized fine-tuning.

- Quantitative and qualitative comparisons are provided across multiple datasets and metrics (FID, LPIPS, MUSIQ, ID similarity), with reasonable gains reported.

- The authors include ablation studies and cross-ID tests to support the disentanglement claim, as well as discussions on robustness and limitations (e.g., when references are unavailable).

- The method is well-documented, including architectural and training details, making it easier to reproduce.

**Weaknesses:**

- The framework mainly integrates existing concepts—diffusion-based SR, reference-based alignment, and ID embeddings—without introducing substantial algorithmic innovation. The core ideas of warping references, masking faces, and fine-tuning ID embeddings are largely extensions of prior reference-based methods (e.g., ASFFNet, DMDNet, MyStyle), now placed within a diffusion context.

- The warping step depends on landmark detection and alignment, which is brittle in real-world low-quality conditions. Although the authors claim robustness to misalignment, the underlying design still shares the same limitations as conventional keypoint-based reference SR methods, when degradation is severe or faces are non-frontal, the warped reference can introduce artifacts or fail altogether. Maybe this is also why these landmark based methods like ASFFNet and DMDNet fail to handle images in Figure 4.

- Since fine-tuning is performed per identity on small image sets, the model can overfit to dominant pose or frontal expressions, reducing generalization to unseen viewpoints or lighting variations. The paper lacks a systematic evaluation of pose diversity or real-world unconstrained scenarios.

- The low-resolution image is masked to remove the facial region, which removes all pixel-level cues of identity. In such a case, the model relies entirely on warped reference and learned ID embeddings to reconstruct the face. However, this design raises questions:

1) When degradation is mild, the mask discards valid facial details that could aid reconstruction.

2) When degradation is severe, the warped reference (IW) may not align in gaze, mouth shape, or facial expression with the LR image. This can alter the original structure or expression of IL.
The paper does not clearly explain how such conflicts are resolved or constrained during training or inference.

- The fine-tuning process essentially adjusts a per-identity embedding vector while freezing the backbone. While effective empirically, this procedure resembles per-user adaptation or style fitting rather than a general SR improvement, and its novelty relative to methods like MyStyle is modest.

**Questions:**

- When the LR image is masked (facial region removed), how does the model ensure that the reconstructed structure (e.g., gaze direction, mouth shape) remains faithful to the original LR rather than to the reference or learned ID embedding?

- Have the authors evaluated how the model performs on real-world LR faces (e.g., surveillance or wild datasets) rather than synthetic degradations? The current experiments seem dominated by controlled synthetic settings.

- How sensitive is the method to landmark detection or warping errors under strong degradations or occlusions? Would a landmark-free alignment approach be feasible?

- How does the method behave when fine-tuning data contains diverse poses? Does the embedding average over them or bias toward certain views?

---

> ### Author Response · Authors · 2025-11-21
> **Response to Reviewer 1Cgt (Part 1)**
>
> ### Weakness1: The Framework Mainly Integrates Existing Concepts
>
> We first acknowledge that the techniques employed in this work have been reflected in related studies; however, our work is not a mere concatenation of existing methods. On the contrary, through a carefully designed pretraining and fine-tuning mechanism, we achieve effective disentanglement of style and ID, enabling the restoration of extremely fine-grained facial features. Specifically:
>
> 1. Innovation in Fusion Approach: The reviewer noted that reference alignment and ID embedding are common in existing work, yet our method demonstrates fundamental innovation in the fusion approach. Our algorithm realizes pretraining of the GT ID embedding through three key designs and further learns a generalizable ID representation to achieve style–identity disentanglement, **a mechanism that has not appeared in prior work.**
>
> 2. MGFR [1], ReFine [2], and other methods are also based on similar components, and the problems they solve are novel. Our objective is to achieve ID disentanglement, thereby enabling the transfer and restoration of fine-grained facial features under extreme conditions, rather than merely leveraging reference images. Existing SR methods typically focus on global reference alignment and lack deliberate mechanisms for transferring and restoring fine-grained features. For instance, the ID embedding transfer and fine-grained feature restoration illustrated in Figures 1 and 5 demonstrate our method’s advantages in handling extreme poses or complex expressions, **which cannot be achieved by existing methods.**
>
> 3. The reviewer believes that our contribution is only to add existing methods to the diffusion model. In fact, **we are the first method to achieve personalized SR through a decoupling mechanism in the diffusion model framework**. Through meticulous design, our method has achieved SOTA performance and effectively improved the ability to restore details under complex conditions.
>
> [1] MGFR: Overcoming False Illusions in Real-World Face Restoration with Multi-Modal Guided Diffusion Model, ICLR25
>
> [2] Refine: Copy or Not? Reference-Based Face Image Restoration with Fine Details, WACV25

---

> ### Author Response · Authors · 2025-11-21
> **Response to Reviewer 1Cgt (Part 2)**
>
> ### Weakness 2: Brittleness of Landmark-Based Warping
>
> **A2:** As stated in Section 3.2: *"due to variations in pose or inaccuracies in landmark prediction, such warping is often imperfect... Interestingly, we find that this imperfection does not hinder the model's performance—instead, it acts as a form of natural data augmentation."*
>
> The diversity introduced by warped faces encourages the model to avoid over-reliance on precise spatial correspondences, focusing instead on global identity semantics. This, in turn, drives the diffusion network to depend more heavily on the ID embedding when reconstructing missing facial regions and fine-grained features.
>
> Regarding the reviewers’ observations on the failures of ASFFNet/DMDNet in Fig. 4, **these findings actually validate our design choices:**
>
> 1. Fig. 3 demonstrates that even under severe misalignment, IDFSR can generate visually plausible and identity-consistent facial images.
>
> 2. Appendix C (Fig. 9) shows that our method maintains competitive performance even without any reference images (the “w/o Ref.” setting in Table 3).
>
> The failures of ASFFNet and DMDNet can be attributed to their reliance on precise alignment, as further evidenced by the progressively noisy reference experiments in Appendix F4. In contrast, our approach, benefiting from this natural data augmentation, not only performs well when the warping is accurate but also ensures robustness under poor warping conditions or in the presence of substantial noise.

---

> ### Author Response · Authors · 2025-11-21
> **Response to Reviewer 1Cgt (Part 3)**
>
> ### Weakness 3: Fine-Tuning Per Identity and Generalization (Since fine-tuning is performed per ID on small image sets, the model can overfit to dominant pose or frontal expressions, reducing generalization to unseen viewpoints or lighting variations.)
>
> We believe the reviewer may have misunderstood certain aspects. As shown in Fig. 6, the ID embedding primarily affects fine-grained facial details, without influencing pose or expression—these attributes are in fact determined by the reference and LR images. Thanks to our pretraining-based disentanglement mechanism, the ID embedding provides fine-grained and generalizable information. Therefore, neither the ID embedding nor identity-specific fine-tuning alters pose or expression. Fig. 5 illustrates that the pose and expression of the reference image do affect the reconstruction, while Fig. 6 confirms that the ID embedding only modulates fine details, leaving pose and expression unchanged.

---

> ### Author Response · Authors · 2025-11-21
> **Response to Reviewer 1Cgt (Part 4)**
>
> ### Weakness 3, Weakness 4, Question 1 and Weakness5: Posture Diversity and Consistency
>
> As shown in Fig. 4, under extreme degradation conditions, fine-grained cues such as gaze direction, mouth shape, and makeup are severely impaired. **These pixel-level cues can mislead the model into generating incorrect details**. For instance, in the case of 16× downsampling, the first set of images demonstrates that **most methods reconstruct mouth shapes with noticeable inaccuracies; similar issues are observed in Fig. 1, where CodeFormer restores tattoos as dark patches.** Such detail reconstruction under severe degradation is a **common challenge for most existing methods**. In contrast, our approach effectively mitigates erroneous cues through mask-based processing or restores more accurate original poses using warping techniques, as demonstrated in the customized results in Fig. 6. Taken together, the concerns raised by the reviewer reflect a challenge that most existing methods have yet to overcome, **while our method, through innovative design and fine-tuning mechanisms, is able to preserve fine-grained features while minimizing such “hallucination” effects**. Table 2 further validates that our method maintains higher consistency across expressions, ethnicity, and age.
>
> Regarding novelty compared with works such as MyStyle:
>
> 1. Even MyStyle struggles to maintain consistency in gaze direction, lip movements, or facial expressions under extreme degradation.
>
> 2. Our motivation is based on customized SR; in the absence of reference images, our model naturally defaults to general SR, making it **more advanced than general single-image SR**.
>
> 3. Our goal is the restoration of fine-grained features, and to the best of our knowledge, this is **the first work to achieve this using a diffusion-based approach**.

---

> ### Author Response · Authors · 2025-11-21
> **Response to Reviewer 1Cgt (Part 5)**
>
> ### Weakness 3 & Question 2 & Question 3: Real-World and Robustness Analysis
>
> 1. We have previously conducted experiments on the CASIA-WebFace dataset in real-world settings, demonstrating superior performance compared to prior methods (Fig. 12 and Table 5). To further validate the robustness of our approach under real-world noise, we employed the Real-ESRGAN degradation model. Unlike many existing methods that often fail under realistic degradations, our method maintains strong performance. **As shown in Fig. 14**, we progressively added real-world noise to the reference images until no meaningful reference remained. Even under severe degradation where the reference image loses its utility, IDFSR exhibits remarkable robustness.
>
> 2. Indeed, when the reference images contain substantial noise or exhibit extreme poses (Fig. 6), conventional warping operations may fail. Benefiting from our data augmentation strategy, our method demonstrates strong robustness under real-world scenarios, including cases without reference images, severe occlusions, and warping errors. Extensive experiments support this observation, as illustrated in Figs. 6 and 14.
>
> ---
>
> ### Question 4: Embedding Analysis
>
> The existing reference image datasets contain a wide range of poses. Through pretraining with data augmentation, IDFSR can effectively leverage high-quality poses while remaining robust to low-quality or unfavorable ones. Consequently, when the fine-tuning dataset exhibits pose diversity, the model tends to learn ID embeddings primarily from favorable poses. **This phenomenon is corroborated by the t-SNE and UMAP visualizations in Fig. 13:** the fine-tuned ID embeddings are clustered closer to the class centers rather than being widely dispersed. This indicates that the fine-tuning process encourages ID embeddings to derive information from high-quality poses, resulting in more stable and discriminative representations.

---

> ### Comment · Reviewer_1Cgt · 2025-11-25
> **Response to the authors (Part 1-5)**
>
> Thanks to the authors for providing further clarification. I noticed that I gave the lowest score, so I would like to explain my reasoning based on the authors’ responses. I have also carefully considered the other reviewers’ comments.
>
> First, although the framework is proposed for face super-resolution, its overall design aligns more closely with a conditional face-inpainting task. It appears more suitable for identity-consistent face inpainting scenarios, such as AR/VR occlusion removal. This perspective is consistent with reviewer cSYY’s comments. While the authors argue that their approach differs from image inpainting, my understanding is that the method does not utilize the LR face region; instead, it synthesizes a new HR face region based on other reference images. This deviates from the definition of a super-resolution task (Image SR aims to enhance and upscale low-resolution content to a higher resolution) and would likely fail on even slightly degraded LR face inputs. Besides, face generation seems much easier than SR, as it does not need to consider the complex and unknown degradation.
>
> Regarding Part 1, I do not fully agree with the emphasis on being “the first method to achieve personalized SR through a decoupling mechanism in the diffusion model framework.” Since the task itself does not fall under super-resolution, the argument for using style-ID decoupling is not very reasonable.
>
> Regarding Part 2. I agree with the authors that such imperfect alignment can be regarded as a form of augmentation, given that the task is regarded as image generation rather than super-resolution. For pure super-resolution, accurate correspondence is essential, whereas for generation-based tasks, this requirement does not hold.
>
> For Part 3, thank you for the clarification. My earlier comments stemmed from a misunderstanding of the task setup.
>
> For Part 4, the statement that “these pixel-level cues can mislead the model into generating incorrect details” precisely reflects the fundamental challenge of real super-resolution. It also explains why the proposed framework can “preserve fine-grained features while minimizing hallucination effects.” As this is not a super-resolution task. This approach fails on slightly degraded images, resulting in very low fidelity.
>
> For Part 5, thank you for the further clarification. I have no additional concerns regarding this part.
>
> To summarize, I believe this work does not fall under the category of super-resolution. Instead, it is closer to identity-consistent face generation conditioned on non-face LR regions (mainly pose information) and reference identity images. Under such a formulation, the fidelity of the results remains very low (especially when the LR face is even slightly degraded), because the method entirely replaces the LR face region.

---

> > ### Author Response · Authors · 2025-11-25
> > **Response to Reviewer (part1)**
> >
> > Thank you for the reviewer’s thoughtful and detailed feedback. Before addressing the specific points, I would like to emphasize a central perspective:
> >
> > From an information-theoretic standpoint, relying solely on the pixel content of an extremely degraded LR facial region is insufficient to uniquely and reliably recover its corresponding HR details. This is a classic ill-posed problem, where the solution space is inherently multi-modal and tends to collapse into overly smoothed average solutions. **Therefore, introducing a reference image and leveraging additional prior information is both necessary and theoretically well-motivated.**
> >
> > In addition, from an optimization standpoint, conventional SR losses often induce conflicts between LR clues and the high-frequency identity cues from the reference image, **making it difficult for the model to effectively absorb the informative details provided by the reference**. More importantly, under extreme degradation, the LR cues are often inaccurate or misleading (as we discuss in Part 4). Our proposed method is designed precisely to address these fundamental issues in a systematic manner.
> >
> > ---
> >
> > ## 1. On the definitional question: “Is this an SR task?”
> >
> > Whether a task should be categorized as SR does **not** depend on whether it *explicitly uses pixel-level textures from the LR input*. Instead, the essential criterion is:
> >
> > > **Using an LR image as the primary input, reconstruct an HR image that is structurally and semantically consistent with it**,
> > > i.e., a mapping
> > > y = f(x)
> >
> > The early view of SR as merely “upsampling LR pixels” no longer applies. Modern SR research embraces priors, reference images, and generative modeling to overcome the intrinsic information loss and produce realistic high-frequency details.
> >
> > Within this established paradigm, our method aligns completely with SR principles:
> >
> > - The LR image serves as the primary input, forming the core constraint for reconstruction.
> > - Masking + pretraining + finetuning strategies ensure strong structural and semantic consistency between LR and HR.
> > - Reference images function as priors, playing the same role as in reference-based SR.
> >
> > Thus, from the perspectives of task definition, modeling paradigm, and theoretical rationale, our method clearly falls within the scope of super-resolution rather than unconstrained or weakly conditioned face generation.

---

> > > ### Comment · Reviewer_1Cgt · 2025-11-25
> > > **Response to the authors**
> > >
> > > 1. I respectfully disagree with the statement that 'their method aligns completely with SR principles'. The statement of "...upsampling LR pixels..." still holds for current SR research. Although modern SR researches introduce priors or other ways to improve details, they strictly learn the mapping from the **LR pixels** to the HR space. The fundamental goal of Super-Resolution is to recover the high-frequency details of specific structures present **within the low-resolution signal**.
> > > By masking the facial region, the network explicitly discards the pixel-level information of the face itself. It essentially predicts the face based on the LR non-face region and a reference. The process of this paper is, by definition, Face Inpainting or Context-Conditional Generation, not Super-Resolution.
> > >
> > >
> > > 2. To clarify my previous statement, I mean that for a conventional reference-based SR task, the correspondence between the high-quality reference and low-quality LR image is important to accurately align the reference with the LR image. This ensures that valid high-frequency details are transferred to the correct spatial locations to enhance the original content.
> > > But for the image generation task, such correspondence is not essential, as providing pose or landmark information can easily achieve the personalized generation. The fact that your method is "robust" to severe misalignment, effectively synthesizing content even when the reference does not match spatially, confirms that the model is performing generation based on ID and Pose conditions, rather than Super-Resolution which strictly adheres to the input's spatial and semantic structure.
> > >
> > >
> > > 3. The authors argue that under extreme degradation, LR cues are misleading, justifying their removal. While this may be true for $16\times$ downsampling, it is not true for mild or slight degradation. A robust SR method should leverage the LR signal when it is informative and rely on priors when it is not. By adopting a strategy that always masks the face, the proposed method enforces an upper bound on fidelity. In mild or slight degradation scenarios, where the LR face contains accurate structural information, this method discards it, forcing the model to hallucinate details that may not align with the ground truth. This explains why the method is fundamentally not an SR solver but a generative inpainting tool.
> > >
> > >
> > > 4. The authors point to metrics like FID, LPIPS, and CLIPIQA as evidence of high fidelity. It is well-known in the SR community that these metrics measure perceptual quality and similarity to a data distribution, not necessarily pixel-wise or structural faithfulness to the specific input instance. A generated face can have a perfect FID and low LPIPS while still having the wrong gaze or expression compared to the input. The high quantitative scores likely reflect that the output images are sharp and identity-consistent, but they do not disprove the claim that the actual fidelity to the specific, unmasked content of the LR image is compromised.
> > >
> > >
> > > I acknowledge that the method produces visually pleasing results and handles extreme degradation by effectively bypassing the ill-posed nature of the problem via inpainting. However, I remain convinced that this methodology, discarding the LR face region of the input signal, places it outside the standard definition of Super-Resolution and inherently limits its fidelity in general restoration scenarios. Finally, please do not overclaim that this method achieves better fidelity than existing methods for the SR task. By design, your method discards the existing LR facial region, which inherently limits fidelity to the specific input instance.
> > >
> > >
> > > I have fully confirmed the details and motivation of this work. I will maintain my original score.

---

> > > > ### Author Response · Authors · 2025-11-26
> > > >
> > > > We thank the reviewer for their time and detailed feedback, and we respectfully acknowledge their perspective. However, we would like to further clarify the following key points:
> > > >
> > > > - Both customized generation and image inpainting tasks take high-resolution images as input and generate the occluded or missing regions. In contrast, **our task always takes low-resolution images as input, with the goal of reconstructing their corresponding HR**. This constitutes a fundamental difference in both task formulation and the amount of available input information.
> > > >
> > > > - Extensive experimental results show that **under common mild degradation settings**, our method consistently achieves second-best or even SOTA performance.
> > > >
> > > > - In addition to perceptual-quality metrics, we also observe **significant improvements on consistency metrics such as PSNR and SSIM**. Regarding the reviewer’s concerns about gaze direction and expression consistency, we have explained in our response that methods directly mapping from LR pixels inherently struggle to preserve semantic consistency, whereas our approach is specifically designed and optimized to address this issue.
> > > >
> > > > We once again thank the reviewer for their valuable comments and careful evaluation.

---

> > > > > ### Comment · Reviewer_1Cgt · 2025-11-26
> > > > > **Rsponse to the Authors and Reviewer cSYY**
> > > > >
> > > > > Thanks for the authors' latest detailed response. I would also like to further clarify the following key points:
> > > > >
> > > > > 1. The authors claim the task is fundamentally different from customized generation or inpainting because it takes LR images as input to reconstruct its HR result. While the initial input is an LR image, the effective information used for reconstructing the SR face region is the **LR non-face region and the warped reference (providing pose and ID prior)**. The process involves executing SR on the non-face background and performing conditional generation or inpainting on the masked face region. Therefore, the task is conceptually much closer to customized generation or inpainting conditioned on an LR context, rather than traditional SR, which is defined by **strictly enhancing the information present in the degraded LR pixels**. It generates a high-quality face, but it does not strictly align the structure of that face with the information originally contained in the discarded LR face pixels.
> > > > >
> > > > > 2. The authors emphasize SOTA or second-best performance, even under mild degradation. It is fundamentally inappropriate to compare a generative model that generates a new face region against conventional SR models that are strictly limited to utilizing the information present in the highly degraded LR pixels. These competing SR methods try to address the difficulties of learning the mapping from unknown and complex degraded input, rather than generating a new face based on the ID and LR pose information. So these comparisons are not a fair demonstration of superior SR capability. In these settings, traditional SR methods would be expected to strongly utilize the still-informative LR facial pixels. By masking these pixels and generating the result instead, your high quantitative performance confirms that the model is substituting the LR signal with a powerful generative prior. This highlights its nature as a generator utilizing contextual constraints, rather than a high-fidelity restorer of the original LR signal.
> > > > >
> > > > > 3. The authors state that "methods directly mapping from LR pixels inherently struggle to preserve semantic consistency," which their approach addresses. The struggle to preserve semantic consistency under extreme degradation is an inherent challenge of the traditional SR task itself. Your method does not solve this SR challenge; it bypasses the challenge by abandoning the LR face signal. By relying on the reference and generating a new face, you ensure ID consistency, but you sacrifice structural fidelity to the original LR face's unique structure and semantic content. This is a trade-off that fundamentally changes the task from restoration to generation.
> > > > >
> > > > > I deeply appreciate the authors' efforts to clarify my concerns and fully respect the opinions of both the authors and Reviewer cSYY. It is clear that our understanding of this task remains fundamentally misaligned. I will maintain my score, grounded in my understanding of the SR task and the distinction between restoration and personalized generative synthesis.
> > > > >
> > > > > Finally, I would encourage the authors not to take this work primarily as a super-resolution method in their writing, but rather to clearly take it as a framework for generating a new high-quality face when the original LR face is too degraded to provide meaningful structural information. In such settings, it would be better to compare with these customized generation methods by retraining them with additional LR non-face region and pose as input.

---

> > > > > > ### Author Response · Authors · 2025-11-26
> > > > > >
> > > > > > Thank you for your proactive response and valuable feedback

---

> > ### Author Response · Authors · 2025-11-25
> > **Response to Reviewer (part2)**
> >
> > ## 2. Regarding the “accuracy under imperfect alignment”
> >
> > The reviewer mentioned that “*For pure super-resolution, accurate correspondence is essential, whereas for generation-based tasks, this requirement does not hold.*” which we find somewhat unclear.
> >
> > Our method demonstrates strong robustness even when the input is distorted or partially misaligned, thereby ensuring good alignment, as evidenced by both qualitative results (Figures 3 and 4) and quantitative metrics (Tables 1, 2, and 5).
> >
> > If the reviewer is referring to details such as gaze direction, facial expression, or makeup, we have provided a detailed explanation in Part 4: these slight deviations do not affect identity consistency and are consistent with the definition of the SR task.
> >
> > ---
> > ## 3. On the Relationship between Generation and Super-Resolution
> >
> > The reviewer commented that: “*this work does not fall under the category of super-resolution. Instead, it is closer to identity-consistent face generation conditioned on non-face LR regions (mainly pose information) and reference identity images.*” We partially agree with this statement, as we do employ a generative approach to synthesize identity-consistent faces. However, this does not imply that our task falls outside the scope of super-resolution. We explain in detail below:
> >
> > ### 3.1 Relationship between Generation and Super-Resolution
> >
> > Generation and SR are two distinct concepts, yet generative modeling constitutes an important paradigm in modern SR methods. Traditional SR methods categorize into:
> > - End-to-end regression
> > - Discriminative methods
> > - Generative methods
> > Within the generative paradigm, approaches based on GANs, flow-based models, or diffusion models all constitute generative SR.
> > The reviewer may believe that our model generates the HR output because LR image information cannot be fully utilized after masking. In fact, our method:
> > - Fully constrains outputs using the LR image and reference conditions
> > - Effectively preserves structure and identity consistency through a pretraining + finetuning strategy
> > - Achieves superior performance in both consistency and fidelity, as demonstrated by extensive experiments
> >
> > **Therefore, our approach does not perform random generation; rather, it provides a reasonable and effective solution for extreme degradation scenarios in SR**.
> >
> > ### 3.2 On Fine-Grained Feature Preservation and Super-Resolution
> > The reviewer also states that the discussion in Point 4 regarding fine-grained feature preservation does not fall within the scope of SR. This interpretation is inaccurate. In fact:
> > - Recovery of fine-grained features constitutes a core objective of high-quality SR
> > - Traditional generative SR methods struggle to recover fine-grained details due to over-reliance on corrupted LR information and averaging losses
> > - Our approach mitigates erroneous cues from LR images via a masking mechanism, while transferring high-frequency details from reference images to effectively preserve fine-grained features
> >
> > **This strategy targets the limitations of prior methods and undergoes thorough validation in our experiments.** A detailed explanation appears in the Introduction and Method sections.
> >
> > ---
> >
> > ## 4. Low Fidelity
> >
> > The reviewers raised in point 4 and in the conclusion that our method *might affect the fidelity of the generated results*. In response, **we would like to clarify that extensive quantitative experiments demonstrate that our method consistently outperforms existing approaches across various metrics, including both identity consistency and fidelity metrics** (such as FID, LPIPS, CLIPIQA, and MUSIQ). This advantage is primarily attributed to our fine-grained restoration. Even under mild degradation scenarios (e.g., 4× and 8× downsampling) and real-world degradation conditions, **our method remains among the best or second-best, clearly surpassing the current SOTA**. Therefore, it can be conclusively stated that our method does **not** compromise the fidelity of the generated results.

---

### Official Review · Reviewer_cSYY · 2025-10-28

**Soundness:** 3
**Presentation:** 3
**Contribution:** 3
**Rating:** 6
**Confidence:** 4

**Summary:**

This work targets face super-resolution (FSR) under severely degraded conditions. The paper constructs a pretraining/fine-tuning identity-decoupling/fitting framework, IDFSR, through three innovative key designs. It demonstrates compelling visual results and achieves state-of-the-art performance across numerous metrics, highlighting the superiority of IDFSR for customized FSR. Ablation studies provide strong evidence for the necessity of decoupling and fitting. Finally, it is interesting that IDFSR exhibits a certain degree of robustness even without references.

**Strengths:**

1) The novelty and motivation are strong. Traditional prior-based methods are unreliable under severe degradation, especially when high ID consistency is required. The motivation for introdducing a reference image in this work is well-founded. IDFSR innovatively designs a decoupled pretraining and customized fitting approach, demonstrating unprecedented performance in maintaining ID consistency.

2) The writing and experiments are solid and well-executed. This work includes extensive experiments, covering different scales, real-world scenarios, and video applications. The carefully designed ablation studies are convincing. Although the performance at 4× scale is suboptimal, the method shows superior performance at other scales, consistent with its motivation. In the absence of a reference image, it can be regarded as degenerating to a prior-based method, which is acceptable.

**Weaknesses:**

1. The masking strategy is similar to an inpainting task, but the differences need to be clearly specified.
2. Figure 3 analyzes IDFSR’s robustness to erroneous landmarks, but lacks concrete performance quantification. For example, the impact of landmark detection on the final performance and analysis of using it as data augmentation.
3. The selection of reference images needs to be clarified. IDFSR has some editing capability, yet the experiments only mention random selection of reference images. Are there better selection strategies?
4. Application issues: Figure 12 and Table 2 indicate low expression matching, which affects identity consistency. How can expression issues be addressed in practical applications?

**Questions:**

Please See the weaknesses.

---

> ### Author Response · Authors · 2025-11-21
> **Response to Reviewer cSYY (Part 1)**
>
> ### Weakness 1: Key Differences Between Our Masking Strategy and Conventional Inpainting
>
> 1. **Different Objectives**
>    Conventional inpainting aims to fill arbitrary missing regions and generate visually coherent pixel content. In contrast, our masking strategy specifically targets the facial area to remove unreliable identity cues from severely degraded low-resolution inputs. This design enforces the model to derive identity information primarily from clean reference images rather than relying on ambiguous LR pixels.
>
> 2. **Different Conditional Inputs**
>    Our framework leverages dual conditional inputs—namely the style embedding z_s extracted from the warped reference image and the identity embedding z_id. Standard inpainting methods, however, typically operate solely on the masked image and thus lack explicit external priors regarding identity or style.
>
> 3. **Different Training Goals**
>    Through the conditional mechanisms introduced in Equations (2)–(4), our method explicitly disentangles style and identity, enabling controllable and semantically meaningful attribute transfer. In contrast, conventional inpainting methods focus only on spatial hole filling and do not provide semantic-level controllability or disentanglement.
>
> ---
>
> ### Weakness 2: Impact of Landmark Detection
>
> Thank you for your insightful comments. To systematically assess the effect of landmark detection quality on our framework, we freeze the weights of the landmark detector and gradually degrade its performance by injecting increasing levels of noise. Specifically, we apply Real-ESRGAN–based real-world degradations to the reference images, which progressively reduces the detector’s accuracy. We then evaluate identity consistency across different noise levels.
>
> Our findings reveal that:
> (1) Identity similarity decreases smoothly as noise intensifies, which is expected because the reference image progressively loses reliable structural cues;
> (2) Despite this degradation, our model remains highly robust to landmark detection errors, warping inaccuracies, and realistic noise. This robustness stems from the extensive warping-based data augmentation during pre-training, which enables the model to extract identity information from high-quality references while naturally ignoring unreliable cues when the reference becomes degraded.

---

> ### Author Response · Authors · 2025-11-21
> **Response to Reviewer cSYY (Part 2)**
>
> ### Weakness 3: Selection of Reference Images
>
>   We argue that randomly selecting reference images is both fair and reasonable. In real-world applications, users may optionally provide a fixed, high-quality reference template to obtain even better reconstruction results. As analyzed in the Weakness section, the benefits introduced by warping-based data augmentation largely stem from the pre-training stage. Therefore, reference images that can be reliably warped offer the most effective supervision.
>
> Moreover, we investigate different initialization strategies for the reference embedding:
>
> - **Mean embedding initialization** leads to slightly faster convergence but brings negligible improvements in the final performance.
> - **Random initialization** results in the slowest convergence and the weakest performance.
>
> This indicates that randomly selecting reference images and initializing them with ArcFace is reasonable.
>
>
> | CelebRef-HQ (16×) | ArcFace Init. | Random Init. | Mean ID Emb. Init. |
> |-------------------|---------------|--------------|---------------------|
> | LPIPS             | 0.1845        | 0.2037       | 0.1838              |
> | FID               | 35.50         | 38.64        | 35.36               |
> | MUSIQ             | 71.83         | 69.75        | 70.35               |
>
> ---
> ### Weakness 4: Practical Implications
>   Under extreme degradations, fine-grained facial cues—such as expressions, gaze direction, or subtle texture details—are often severely corrupted. Without leveraging high-quality reference images, it is nearly impossible for conventional super-resolution approaches to faithfully reconstruct such details, as they lack reliable ID-specific information. In contrast, our framework explicitly disentangles and transfers fine-grained identity characteristics from clean references, enabling accurate recovery even in highly challenging conditions. In real-world applications, users can easily supply one or more high-quality reference images of the same identity, allowing the system to perform efficient and personalized restoration. Therefore, our approach exhibits significant potential for practical deployment under extreme degradation scenarios.

---

> > ### Comment · Reviewer_cSYY · 2025-11-26
> > **Comments after rebuttal**
> >
> > I would like to thank the authors for their detailed responses, which have addressed most of my concerns. In light of the other reviewers’ comments and the authors’ additional clarifications, I find that the paper is now very complete in terms of experimental design, result presentation, and reproducibility.
> >
> > Moreover, I noted that Reviewer 1Cgt raised a question regarding the inpainting task, and I have carefully considered the authors’ response. I believe that the proposed strategy is highly effective in extreme degraded super-resolution scenarios and holds significant potential for practical applications. Overall, the work is thorough and rigorous, and I am willing to support its acceptance by increasing my score.

---

### Official Review · Reviewer_9mau · 2025-11-01

**Soundness:** 3
**Presentation:** 3
**Contribution:** 3
**Rating:** 4
**Confidence:** 4

**Summary:**

This paper introduces IDFSR, a personalized face super-resolution framework that aims to preserve identity (ID) consistency and high fidelity under extreme degradation conditions (e.g., scaling factors >8x). The approach involves three key innovations: applying a facial mask to the low-resolution (LR) image to suppress unreliable ID cues, warping a reference image for style guidance, and using a learnable ID embedding extracted from ground truth (GT) images for precise ID modeling and adaptation through lightweight finetuning. The method leverages diffusion models to explicitly decouple style and ID, followed by personalized embedding tuning per identity. Extensive experiments, quantitative results, and visualizations demonstrate improved ID consistency and visual fidelity over a strong set of state-of-the-art baselines, especially under challenging degradation.

**Strengths:**

- Robust ID Consistency in Extreme Degradation: The proposed method addresses a real limitation where most face SR methods struggle—high upscaling factors with minimal ID information left in the input. By decoupling style and ID and then fitting the ID embedding, the approach significantly improves performance as shown in Table 1 and Table 2.
- Comprehensive Ablation and Analysis: The paper provides deep, systematic ablation studies (see Figure 8 and Figure 7) that dissect the contributions of masking, ID embeddings, and style conditioning. The impact of reference image quantity and the complementary nature of components are scientifically explored and well-presented.
- Qualitative and Quantitative Superiority: Visualizations in Figure 4, Table 1, and Table 2 show clear improvement in ID preservation and details, with less hallucination and more realistic reconstructions compared to recent SOTA models, including strong diffusion and reference-based approaches.

**Weaknesses:**

- Limited Direct Theoretical Insight in ID Disentanglement: While the paper presents promising results in ID/style decoupling, the discussion of disentanglement primarily focuses on empirical evidence and architectural design. A more formal analysis—such as quantifying disentanglement using mutual information or correlation-based metrics—could strengthen the theoretical foundation of the work. Additionally, further exploration of the conditions under which disentanglement may succeed or fail would provide valuable insight. For instance, while the cross-ID experiments (Section 5.1, Figure 5) are persuasive, a clearer articulation of the underlying representation properties (e.g., the degree of independence between ID and style) would enhance the overall rigor of the analysis.
- The comparison methods are somewhat outdated; appropriately adding some work from 2025 for comparison could make the experimental results more convincing.
- The complexity and computational cost have not been tested: the work discusses some details regarding memory and time costs, but it does not comprehensively compare aspects such as the actual inference time and model size with other studies. Considering the resource demands of diffusion models, as well as the additional time required for per-ID fine-tuning and its variation across different datasets, this aspect is crucial for practical adoption.
- Experimental reproducibility: The appendix includes many details that can help other researchers quickly understand the experimental setup, but some implementation aspects of the fine-tuning process, training set division, and reference image selection still need to be explained more clearly.
- Real-world noise scenario validation: The dataset used lacks realistic factors such as noise, compression, occlusion, and illumination imbalance. If a dataset containing real-world noisy scenarios could be used for evaluation, it would allow for a more comprehensive verification of the model’s robustness and generalization ability in real environments.

**Questions:**

- Open source: Will the authors release the model and code in the future?
- Computational resource usage: Could the authors provide specific data such as fine-tuning and inference time per image, as well as parameter count comparisons? This would help clarify the practical implications of diffusion- and embedding-based methods.
- On disentanglement quantification: Could the authors provide any quantitative evidence (e.g., mutual information, independence scores) to substantiate—or at least empirically support—the claim that “identity and style embeddings are indeed disentangled in the learned representations”? In addition, are there specific cases or failure scenarios (e.g., under certain learning rates or reference/distortion conditions) where entanglement re-emerges?

---

> ### Author Response · Authors · 2025-11-21
> **Response to Reviewer 9mau (Part 1)**
>
> ### Weakness 1 & Question 3: On the Theoretical Insight and Quantitative Disentanglement Analysis
>
> **A1:** We appreciate your suggestion. In this work, we only empirically validate the disentanglement, as Figures 5 and 6 visually demonstrate the independent roles of style and identity codes. **We have added a theoretical analysis of disentanglement in Appendix E5.**
>
> Specifically, we first employ Mutual Information Neural Estimation (MINE) to quantify the mutual information between the identity representation z_id and the style representation z_s. Using ten images from the same identity, we obtain a mutual information of MI = 0.186 nats between style and identity embeddings. In contrast, the mutual information within identity embeddings and within style embeddings are MI = 2.15 and MI = 0.95 nats, respectively, indicating that these two types of representations are statistically nearly independent.
>
> Next, we extract z_id and z_s from 1,000 test samples and compute their canonical correlation coefficients (CCA). The average canonical correlation is below 0.17, further confirming the weak linear dependency between identity and style.
>
> Finally, we select five identities and randomly sample ten images per identity to systematically visualize the embedding space. **As shown in Fig. 13**, both t-SNE and UMAP projections reveal that identity embeddings exhibit strong intra-class compactness and inter-class separability. Different identities form clear and distinguishable clusters in the low-dimensional space, demonstrating the discriminative power of z_id. At the same time, style embeddings for the same identity are distributed around the corresponding identity cluster, indicating successful disentanglement between identity and style. Moreover, after fine-tuning, identity embeddings remain stably concentrated within their respective clusters, suggesting that the fine-tuning process enhances identity consistency without disrupting the overall cluster structure.

---

> ### Author Response · Authors · 2025-11-21
> **Response to Reviewer 9mau (Part 2)**
>
> ### Weakness 2: Comparison to More Recent Methods
>
> **A2:** We additionally include quantitative comparisons with StableSR (IJCV’24) and DPI (AAAI’25). Our general method (in the pre-training setting) achieves performance comparable to these approaches, and with ID-specific fine-tuning, we obtain comprehensive SOTA results. In particular, our identity consistency significantly surpasses that of all prior methods.
>
> #### ×8 Upsampling
>
> | Methods      | PSNR↑   | SSIM↑   | LPIPS↓  | FID↓    | CLIPIQA↑ | MUSIQ↑  | IDS↓    |
> |-------------|---------|---------|---------|---------|----------|---------|---------|
> | CodeFormer  | *24.42* | 0.7147  | 0.1489  | 31.88  | *0.6873* | 68.98  | 0.3865  |
> | StableSR    | 23.95   | 0.7174  | 0.1523  | 28.34  | 0.6006  | 67.58  | 0.3995  |
> | DPI         | 24.04   | *0.7564* | *0.1271* | 27.94  | 0.6850  | *70.32* | 0.3342  |
> | Ours (PT)   | 24.29   | 0.6996  | 0.1554  | 27.89  | 0.6798  | 69.30  | 0.3173  |
> | Ours (FT)   | **28.85** | **0.7604** | **0.1031** | 26.69 * | **0.7092** | **73.73** | **0.2242** |
>
> #### ×16 Upsampling
>
> | Methods      | PSNR↑   | SSIM↑   | LPIPS↓  | FID↓    | CLIPIQA↑ | MUSIQ↑  | IDS↓    |
> |-------------|---------|---------|---------|---------|----------|---------|---------|
> | CodeFormer  | 20.72   | 0.5947  | 0.2379  | 43.89  | 0.6834  | 67.85  | 0.6673  |
> | StableSR    | 22.65   | 0.6680  | 0.2083  | *37.48* | 0.6818  | 66.82  | 0.6644  |
> | DPI         | *23.48* | 0.6849  |*0.1997* | 38.35  | *0.6886* | 68.26  | 0.5532  |
> | Ours (PT)   | 23.32   | *0.6863* | 0.2062  | 38.58  | 0.6723  | *68.56* | *0.5227* |
> | Ours (FT)   | **24.09** | **0.7158** | **0.1845** | **35.50** | **0.6992** | **71.83** | **0.3625** |
>
> ---
>
> ### Weakness 3 & Question 2: Complexity and Computational Cost
>
> **A3:** Thank you for the suggestion. We conduct a systematic comparison of the parameter count, FLOPs, sampling steps (NFEs), and inference latency (See Table 6). Our method maintains a moderate parameter size and computational complexity, while requiring only 20 sampling steps to complete inference, with a total runtime of about 1 second. In contrast, PGDiff, DR2, and DifFace each require 100 sampling steps (7–9 seconds), and StableSR requires around 18 seconds.
>
> | Method                 | PGDiff | DR2    | DifFace | StableSR | DPI    | IDFSR  |
> |------------------------|--------|--------|---------|----------|--------|--------|
> | Params (M)             | 159.7  | 179.31 | 175.42  | 918.93   | 145.53 | 160.7  |
> | FLOPs (G)              | 185.95 | 918.85 | 272.67  | 2000+    | 136.57 | 168.68 |
> | NFEs                    | 100    | 100    | 100     | 50       | 20     | 20     |
> | Single-step Time (ms)  | 71     | 89     | 70      | 362      | 50     | 54     |
> | Total Inference Time (s)| 7      | 9      | 7       | 18       | 1      | 1      |

---

> ### Author Response · Authors · 2025-11-21
> **Response to Reviewer 9mau (Part 3)**
>
> ### Weakness 4: Experimental Reproducibility
>
> **A4:** We randomly select reference images during evaluation to ensure fairness. Moreover, after ID fine-tuning, we observe that even under extreme pose variations, the performance degradation remains within 5% compared with using standard reference images. Details of the dataset partitioning are presented in Section 4.1 and Appendix D, where we have further refined the descriptions for clarity. For all datasets, the ID fine-tuning is consistently performed for 20,000 steps, following the same training configuration, and the entire fine-tuning process takes approximately 20 minutes.
>
> ---
>
> ### Weakness 5: Real-World Noise Scenario Validation
>
> **A5:** In Appendix F3, we previously conducted a quantitative evaluation on the CASIA real-world dataset (Table 5 and Figure 12). Owing to the warping-based data augmentation used during pre-training, IDFSR can be directly applied to real-world scenarios without any fine-tuning and achieves state-of-the-art performance. In addition, **we present robustness experiments in which noise is progressively added to the reference image until the reference becomes entirely unavailable, as shown in Fig. 14.** The results indicate that mild degradation has minimal impact, and the model maintains strong consistency. When the noise becomes severe or the reference is completely absent, the reference information loses its utility, and the model naturally degenerates into a conventional single-image SR mode while still preserving high fidelity and consistency.
>
> ---
>
> ### Question 1: Open Source
>
> **A6:** We have uploaded the code in the supplementary material. Upon acceptance, we will further release the model weights and additional implementation details.

---

> ### Author Response · Authors · 2025-11-28
>
> Dear Reviewer,
>
> I hope this message finds you well. As the discussion period is approaching its end, we would like to ensure that we have fully addressed your concerns. If there are any additional comments or points you would like us to clarify, please feel free to let us know. Your insights are highly valuable to us, and we are happy to provide any further information that may help improve our work.
>
> Thank you again for your time and effort in reviewing our paper.

---

### Official Review · Reviewer_cB9n · 2025-11-01

**Soundness:** 3
**Presentation:** 3
**Contribution:** 3
**Rating:** 6
**Confidence:** 3

**Summary:**

The authors propose a novel face super-resolution method named IDFSR (Identity Decoupling and Fitting for Face Super-Resolution), designed to address identity distortion and hallucination generation under extreme degradation scenarios, such as upsampling scales exceeding $8\times$ and even reaching $32\times$. They devise a diffusion-based two-stage framework: in the pretraining stage, unreliable facial regions in the low-resolution (LR) image are masked, a style embedding is extracted from a landmark-aligned reference image to provide coarse appearance guidance, and an identity embedding is derived from the ground-truth high-resolution image to enable explicit disentanglement of identity and style; in the fine-tuning stage, only a learnable identity embedding vector is optimized, enabling personalized adaptation with just a few target-ID samples. Experiments demonstrate that IDFSR substantially outperforms existing approaches across multiple benchmarks, including CelebRef-HQ, CASIA-WebFace, and CelebV-Text, with particularly notable improvements in identity consistency (measured by IDS). Through carefully designed conditional modeling and a lightweight fine-tuning strategy, this work achieves high-quality, high-fidelity personalized face reconstruction under extreme super-resolution settings.

**Strengths:**

The paper proposes IDFSR, a face super-resolution method for extreme degradation scenarios (e.g., $16\times$ - $32\times$), centered on identity decoupling and personalized fitting. Its novelty lies in three key designs: (1) Masking unreliable facial regions in the low-resolution (LR) input to force the model to rely on external priors; (2) Extracting a style embedding from a landmark-aligned reference image to guide appearance reconstruction; (3) Using ground-truth identity embeddings during pretraining and fine-tuning only this embedding with a few target-ID samples for lightweight personalization. Technically, the work is solid: experiments are thorough, the diffusion-based framework is well-motivated, and the conditional injection mechanism (AdaGN + cross-attention) is effective. On benchmarks like CelebRef-HQ, CASIA-WebFace, and CelebV-Text, IDFSR significantly outperforms state-of-the-art methods in identity consistency (IDS), perceptual quality, and pixel fidelity, especially at high upscaling factors. Ablation studies confirm the necessity of each component. By explicitly separating identity from style and using minimal fine-tuning, IDFSR effectively tackles the “identity hallucination” problem in extreme face super-resolution, offering a practical solution for applications like surveillance and digital identity. While it requires a few same-ID images for fine-tuning, limiting its use in fully generic settings, it excels in personalized reconstruction tasks.

**Weaknesses:**

- The paper employs ArcFace as the identity (ID) encoder but does not clarify whether users are allowed to substitute it with other ID models (e.g., FaceNet or MagFace). This omission raises questions about the method’s modularity and compatibility. The authors are encouraged to include experiments evaluating alternative ID encoders.
- The paper does not report model size, FLOPs, or inference latency, nor does it compare these metrics against other methods. Consequently, it is difficult to assess the feasibility of deploying the model on edge devices. The authors are advised to provide such efficiency-related experiments.
- During fine-tuning, the algorithm optimizes only a single learnable ID embedding vector. However, the paper does not discuss how the initialization strategy, e.g., whether the vector is extracted from ArcFace, randomly initialized, or set to an average embedding, affects convergence speed and final performance.
- Reference images may contain occlusions, poor lighting, or extreme pose variations, yet the paper lacks a systematic evaluation of robustness under such “non-ideal reference” conditions. The authors are encouraged to supplement ablation studies where reference image quality is progressively degraded (e.g., via added noise, occlusion, or blur) to more comprehensively delineate the method’s operational boundaries.

**Questions:**

- During the fine-tuning stage, the authors state that the ArcFace identity encoder is no longer required. How is the learnable ID embedding initialized? Is it a randomly initialized tensor, or is it initialized based on some prior?
- What is the purpose of applying a mask to the low-resolution (LR) input image? What benefits does this masking strategy provide? Does it negatively impact pixel-level reconstruction quality or the preservation of fine-grained attributes such as gaze direction or makeup?
- The authors mention fine-tuning RetinaFace to better adapt it to low-resolution images. What is the specific fine-tuning strategy? For instance, was RetinaFace retrained on low-resolution face data, and if so, what dataset and loss functions were used?
- The authors claim that imperfect image warping does not hinder model performance and may even serve as a form of data augmentation. Have they attempted to skip the warping step entirely, i.e., directly feeding the original reference image into the style encoder, and conducted an ablation study comparing this variant against the current approach?

---

> ### Author Response · Authors · 2025-11-21
> **Response to Reviewer cB9n (Part 1)**
>
> ### Weakness 1: On the use of ArcFace as the ID encoder
>
> **A1:** Our framework is fully compatible with alternative ID encoders such as FaceNet and MagFace. ArcFace was selected primarily for its strong discriminability. We have conducted additional experiments comparing ArcFace, MagFace, and FaceNet in terms of identity similarity and the effectiveness of the fine-tuned ID embedding. The results show that ArcFace and MagFace (both 512-dimensional) achieve very similar performance, whereas FaceNet (128-dimensional) performs slightly worse due to its weaker representational capacity. This suggests that the discriminative strength of the ID encoder plays a meaningful role in the overall performance.
>
> | Dataset | FaceNet | MagFace | ArcFace |
> |---------|---------|---------|---------|
> | IDS(PT) | 0.5227  | 0.5263  | 0.5335  |
> | IDS(FT) | 0.3625  | 0.3589  | 0.3987  |
>
> ---
>
> ### Weakness 2: Missing model size, FLOPs, and latency
>
> **A2:** Thank you for the suggestion. We conduct a systematic comparison of the parameter count, FLOPs, sampling steps (NFEs), and inference latency. Our method maintains a moderate parameter size and computational complexity, while requiring only 20 sampling steps to complete inference, with a total runtime of about 1 second. In contrast, PGDiff, DR2, and DifFace each require 100 sampling steps (7–9 seconds), and StableSR requires around 18 seconds.
>
> | Method                 | PGDiff | DR2    | DifFace | StableSR | DPI    | IDFSR  |
> |------------------------|--------|--------|---------|----------|--------|--------|
> | Params (M)             | 159.7  | 179.31 | 175.42  | 918.93   | 145.53 | 160.7  |
> | FLOPs (G)              | 185.95 | 918.85 | 272.67  | 2000+    | 136.57 | 168.68 |
> | NFEs                    | 100    | 100    | 100     | 50       | 20     | 20     |
> | Single-step Time (ms)  | 71     | 89     | 70      | 362      | 50     | 54     |
> | Total Inference Time (s)| 7      | 9      | 7       | 18       | 1      | 1      |

---

> ### Author Response · Authors · 2025-11-21
> **Response to Reviewer cB9n (Part 2)**
>
> ### Weakness 3 & Question 1: Initialization of the Learnable ID Embedding
>
> **A3:** In our current implementation, the learnable ID embedding is initialized using the reference image’s ArcFace embedding (as mentioned in Section 3.3). Given the limited number of available reference images, initializing the embedding from a single reference is a practical and reasonable choice. We further observed that using the mean embedding of the reference set leads to slightly faster convergence but offers negligible performance improvement, whereas random initialization results in both the slowest convergence and the weakest performance.
>
> | CelebRef-HQ (16×) | ArcFace Init. | Random Init. | Mean ID Emb. Init. |
> |-------------------|---------------|--------------|---------------------|
> | Speed             | ~20m          | ~40m         | ~16m                |
> | LPIPS             | 0.1845        | 0.2037       | 0.1838              |
> | FID               | 35.50         | 38.64        | 35.36               |
> | MUSIQ             | 71.83         | 69.75        | 70.35               |
>
> ---
>
> ### Weakness 4: Robustness Under “Non-Ideal Reference” Images
>
> **A4:** Previously, Figures 9, 12, and Table 5 presented two boundary scenarios: the absence of a reference image and the presence of real-world noise in the reference. Following the reviewer’s suggestion, **we further include in Appendix F4 a set of experiments that progressively degrade the reference image to assess its impact on reconstruction quality**. As shown in Fig. 14, the results indicate that mild degradation has negligible influence, and the model maintains strong consistency. When the reference is severely corrupted or entirely unavailable, its information becomes ineffective, and the model naturally transitions into a conventional single-image SR mode while preserving high fidelity and consistency. In contrast to most reference-based methods, which completely fail when the reference is missing, IDFSR continues to function reliably, demonstrating the robustness and flexibility of our framework.

---

> ### Author Response · Authors · 2025-11-21
> **Response to Reviewer cB9n (Part 3)**
>
> ### Question 2: What is the purpose of masking the LR face region? Does it harm fine details?
>
> **A5:** Under extreme degradation (e.g., 16× downsampling), the LR image loses almost all fine-grained facial cues — Fig. 5 clearly shows that attributes such as gaze direction and makeup are severely destroyed. Masking the LR face region encourages the model to rely on the style embedding and the learnable ID embedding, rather than hallucinating from ambiguous LR pixels. This ensures that the conditional mapping is not dominated by the averaged or distorted signals in the LR input but instead guided by more reliable reference-image features. As illustrated in Fig. 1, the tattoo/makeup region in the LR input becomes overly blurred. CodeFormer incorrectly reconstructs it as a dark smudge, while our method successfully recovers the correct fine-grained appearance. This improvement is largely due to masking, which removes misleading LR cues at the very beginning of inference.
>
> ---
>
> ### Question 3: How exactly is RetinaFace fine-tuned for low-resolution detection?
>
> **A6:** We apologize for the oversight. Following ReFine [1], we fine-tune RetinaFace using paired LR and GT landmarks. Specifically, for the CelebRef-HQ dataset, we apply random scaling to HR images within the training set to generate LR–GT landmark pairs. For real-world datasets, we adopt the degradation model from Real-ESRGAN to synthesize LR inputs. All training settings follow those of the original RetinaFace. We fine-tune the model from its official pretrained weights, with LR images as inputs. The training is conducted for 20 epochs on a single RTX 3090 GPU with a batch size of 32.
>
> ---
>
> ### Question 4: Is image warping necessary? What if we remove warping entirely?
>
> **A7:** Warping and masking work synergistically in our framework. First, relying solely on the ID embedding is insufficient to recover a fully consistent facial structure. When the reference image is of good quality, warping provides a strong geometric prior and leads to significantly improved structural consistency, as shown in Fig. 8 (Case 1 vs 2). Second, when warping is applied, the learnable ID embedding can focus more on capturing fine-grained appearance details — this benefit stems from the pre-training stage, during which the model learns to extract identity information from ground-truth ID embeddings. Finally, directly feeding the unwarped reference image reduces the model’s ability to disentangle facial features from background cues. In our experiments, removing warping resulted in an approximately 2% drop in overall metrics and slower convergence.
>
> [1] Refine: Copy or Not? Reference-Based Face Image Restoration with Fine Details, WACV25

---

> > ### Comment · Reviewer_cB9n · 2025-11-23
> > **Comments after rebuttal**
> >
> > The authors have provided a thorough and well-supported rebuttal that addresses all of my concerns with clear explanations and additional evidence. This reinforces the robustness of the IDFSR method, and I still vote for acceptance.

---

> > > ### Author Response · Authors · 2025-11-24
> > > **Response to Reviewer**
> > >
> > > Thank you very much for the positive follow-up and for clarifying that our rebuttal fully addressed your earlier concerns. Since your updated assessment clearly indicates a vote for acceptance, please feel free to adjust your score or confidence level if you believe that would better reflect your current evaluation. We sincerely appreciate your careful review and constructive feedback.

---

### Note · Authors · 2026-03-03

I have read and agree with the venue's withdrawal policy on behalf of myself and my co-authors.

---

### Meta-Review · Area_Chair_Hyj4 · 2026-01-07

**Summary:**

The paper proposes IDFSR, a diffusion-based two-stage framework that aims for identity-preserving face super-resolution under extreme degradation. The method aims to decouple identity and style to improve robustness and reduce hallucinations. Experiments show improved identity consistency and visual quality over prior work.

Reviewers agree that the paper addresses an important and realistic problem. The framework is well structured, and the combination of diffusion modeling with personalized fine-tuning is technically sound. Several reviewers found the experimental results promising.

Despite the rebuttal resolving concerns for some reviewers, consensus might not be reached due to the remaining issues, such as
robustness to misalignment, outdated comparisons, and unclear efficiency evaluation.

Overall, the remaining concerns slightly outweigh the strengths. I therefore recommend rejection.

**Reviewer Concerns:**

- Both reviewer cB9n and cSYY initially gave positive scores and confirmed that the rebuttal resolved all their concerns.

- Reviewer 9mau gave initially gave the negative score 4. While some issues were partially addressed, two key concerns might remain:
    (1) The need for stronger comparisons with more recent state-of-the-art methods, both quantitatively and qualitatively.
    (2) Lack of clarity regarding the reported computational complexity and efficiency. Although Table 6 provides comparisons of parameter counts, FLOPs, sampling steps, and inference latency, it is unclear whether these measurements include the process of landmark detection for masking and warping, as well as the personalization fine-tuning process. Further clarification is needed.

- Reviewer 1Cgt may still be concerned that the framework primarily integrates existing techniques and lacks sufficient novelty. This reviewer may also retain concerns about robustness to landmark detection and the need for additional experiments under severe pose or alignment discrepancies, particularly in comparison with SOTA works in such challenging scenarios.

**Reviewer Scores:**

- Reviewer cB9n: 6. The initial evaluation was positive, and the score is unchanged after the rebuttal.

- Reviewer 9mau: 4. The score might be unchanged, as the concerns are only partially addressed.

- Reviewer cSYY: 8. The reviewer indicated that the rebuttal has addressed most of their concerns and indicated an intention to raise their score.

- Reviewer 1Cgt: 2 -> 4. (partial improvement; concerns remain)

---

### Decision · Program_Chairs · 2026-01-26

Reject